# p38-MK2 signaling axis regulates RNA metabolism after UV-light-induced DNA damage

Marina E. Borisova[1], Andrea Voigt[1], Maxim A.X. Tollenaere[2], Sanjeeb Kumar Sahu[1], Thomas Juretschke[1], Nastasja Kreim[1], Niels Mailand [3], Chunaram Choudhary[4], Simon Bekker-Jensen[2], Masato Akutsu[5], Sebastian A. Wagner [6,7,8] & Petra Beli [1]

Ultraviolet (UV) light radiation induces the formation of bulky photoproducts in the DNA that globally affect transcription and splicing. However, the signaling pathways and mechanisms that link UV-light-induced DNA damage to changes in RNA metabolism remain poorly understood. Here we employ quantitative phosphoproteomics and protein kinase inhibition to provide a systems view on protein phosphorylation patterns induced by UV light and uncover the dependencies of phosphorylation events on the canonical DNA damage signaling by ATM/ATR and the p38 MAP kinase pathway. We identify RNA-binding proteins as primary substrates and 14-3-3 as direct readers of p38-MK2-dependent phosphorylation induced by UV light. Mechanistically, we show that MK2 phosphorylates the RNA-binding subunit of the NELF complex NELFE on Serine 115. NELFE phosphorylation promotes the recruitment of 14-3-3 and rapid dissociation of the NELF complex from chromatin, which is accompanied by RNA polymerase II elongation.

[1] Institute of Molecular Biology (IMB), Ackermannweg 4, 55128 Mainz, Germany. [2] Cellular Stress Signaling Group, Department of Cellular and Molecular Medicine, Center for Healthy Aging, University of Copenhagen, Blegdamsvej 3C, 2200 Copenhagen, Denmark. [3] Ubiquitin Signaling Group, Novo Nordisk Foundation Center for Protein Research, Faculty of Health and Medical Sciences, University of Copenhagen, Blegdamsvej 3B, 2200 Copenhagen, Denmark. [4] Proteomics and Cell Signaling Group, Novo Nordisk Foundation Center for Protein Research, Faculty of Health and Medical Sciences, University of Copenhagen, Blegdamsvej 3B, 2200 Copenhagen, Denmark. [5] Institute of Biochemistry II, Goethe University Medical School, Theodor-Stern-Kai 7, 60590 Frankfurt and Buchmann Institute for Molecular Life Sciences (BMLS), Goethe University, Max-von Laue-Strasse 15, 60438 Frankfurt, Germany. [6] Department of Medicine, Hematology/Oncology, Goethe University, Theodor-Stern-Kai 7, 60590 Frankfurt, Germany. [7] German Cancer Consortium (DKTK), 69120 Heidelberg, Germany. [8] German Cancer Research Center (DKFZ), 69120 Heidelberg, Germany. Correspondence and requests for materials should be addressed to P.B. (email: p.beli@imb-mainz.de)

Ultraviolet (UV) light is a natural source of DNA damage that triggers the formation of cyclobutane–pyrimidine dimers and 6–4 pyrimidine–pyrimidone photoproducts. UV-light-induced DNA damage is recognized and repaired by the components of the global genome or transcription-coupled (TC) nucleotide excision repair (NER) pathway in human cells[1]. The formation of single-stranded DNA during NER and stalled replication forks activate the protein kinase Ataxia telangiectasia and Rad3 related (ATR) and its downstream target Checkpoint kinase 1 (Chk1), which in turn phosphorylate a number of proteins to activate cell cycle checkpoints[2]. In addition to ATR, the mitogen-activated protein kinase 14 (MAPK14, also known as p38 MAPK), is activated in human cells after exposure to UV light[3,4]. The p38 MAPK (of which the α isoform is highly expressed in most human cell types) is a central transducer of cellular stress signaling that is activated by different stress-inducing agents, as well as extracellular signaling molecules such as hormones and cytokines[3,4]. Depending on the stimuli and cell type, upstream kinases MKK3, MKK4, and MKK6 activate p38 by phosphorylation on threonine 180 and tyrosine 182[5]. In response to stress, p38 phosphorylates and activates ~ 10 downstream kinases, including MK2/3/5, MSK1/2, and MNK1/2[3,4]. Recent studies provided evidence for an extensive interplay between UV-light-induced DNA damage and cellular RNA metabolism: exposure of human cells to UV light globally impacts on different RNA metabolic processes, including transcription, splicing, and translation[6–12]. Moreover, components of the transcriptional machinery were shown to be phosphorylated after UV light[13]. Regulation of transcription in response to environmental cues and during development is achieved through the release of paused RNA polymerase II (RNA pol II) from promoter-proximal sites of defined sets of genes[14,15]. UV light has been shown to affect both transcriptional initiation and elongation in human cells[12,16–19]. Although the components and the regulatory mechanisms of DNA repair are relatively well established, the understanding of the signaling pathways and molecular mechanisms that orchestrate the complex changes in transcription and RNA metabolism in general after UV-light-induced DNA damage is only beginning to emerge.

Here we employ quantitative mass spectrometry (MS)-based proteomics to provide a global view on phosphorylation-dependent signaling induced by UV light. We define the cellular phosphorylation events dependent on the ATM/ATR and the p38 MAPK pathway, and determine functional contributions of these pathways to the UV-light-induced DNA damage response. Whereas ATM/ATR primarily phosphorylate proteins involved in DNA repair and cell cycle regulation, the p38-MK2 signaling axis phosphorylates a multitude of RNA-binding proteins. We show that MK2-dependent phosphorylation of cellular proteins triggers the recruitment of 14-3-3 dimers. Mechanistically, we demonstrate that p38-MK2-dependent phosphorylation of the negative elongation factor (NELF) complex promotes its rapid release from chromatin, which correlates with RNA pol II elongation. The presented datasets of p38-MK2/3-dependent phosphorylation sites and 14-3-3 binding proteins will facilitate further studies regarding the roles of p38-MK2/3 signaling axis in the regulation of the cellular RNA metabolism in response to UV light.

## Results

**p38 signaling has a broad regulatory scope after UV light.** We aimed to employ quantitative phosphoproteomics to decipher the signaling downstream of the p38 MAP kinase activated after UV light exposure. We first used western blotting to examine the dynamics of p38 activation after UV light (40 J/m², 1 h recovery)

and found that p38 activation peaked between 30 and 60 min after irradiation, and gradually decreased being almost undetectable 4 h post irradiation (Fig. 1a). Phosphorylation of p38 increased in a dose-dependent manner and was detectable after irradiation of cells with 10 J/m² (Supplementary Figure 1a). In addition to UV light, treatment of cells with UV light mimetic drug 4-NQO and oxidative stress-inducing agent $H_2O_2$ resulted in rapid phosphorylation of p38 (Fig. 1b). In contrast, double-strand DNA break-inducing agents and the replication stress-inducing agent hydroxyurea did not lead to marked activation of p38 early after treatment, although these drugs activated the canonical DNA damage response detected by Chk1 phosphorylation (Fig. 1b). In accordance with previous findings[20], specific inhibition of phosphoinositide 3-kinase-like kinases ATM, ATR, or DNA-PKcs did not affect p38 activation after UV light, suggesting that parallel activation of the ATR-Chk1-dependent canonical DNA damage signaling and p38 MAPK regulates early response of cells to UV light (Supplementary Figure 1b). Chemical inhibition of p38 significantly sensitized U2OS cells to different doses of UV light, indicating that p38 promotes cellular survival after UV light exposure (Supplementary Figure 1c). We performed proteome-wide identification of p38-dependent phosphorylation after UV light to define signaling downstream of p38. To this end, we employed titanium dioxide (TiO2)-based enrichment of phosphorylated peptides followed by peptide identification using ultra-high performance liquid chromatography-tandem MS (LC-MS/MS). Stable isotope labeling with amino acids in cell culture (SILAC) was used to relatively quantify the abundance of the phosphorylated peptides in different experimental conditions. Light-labeled cells were mock-treated and used as control, medium-labeled cells were irradiated with UV light (40 J/m², 1 h recovery), and heavy-labeled cells were pretreated with the specific p38 inhibitor SB203580 followed by irradiation with UV light (Fig. 1c).

We quantified 13,091 phosphorylation sites, of which 10,448 were identified in 2 independent replicate experiments (Fig. 1d and Supplementary Data 1). We observed an excellent quantitative reproducibility between the replicate experiments (Supplementary Figure 1d,e). To determine significantly regulated phosphorylation sites after UV light exposure and sites that are affected by p38 inhibition, we applied a moderated $t$-test (limma algorithm) (Supplementary Figure 1f,g). This analysis revealed that 538 (4.1%) and 153 (1.2%) out of 13,091 phosphorylation sites were significantly upregulated and downregulated, after irradiation of cells with UV light, respectively ($p$-value < 0.01, moderated $t$-test) (Fig. 1d and Supplementary Data 1). Notably, UV-light-induced phosphorylation of 138 phosphorylation sites (25.6%) significantly decreased after p38 inhibition, indicating that phosphorylation of these sites is dependent on p38 activity (Fig. 1d). Transient knockdown of p38 using small interfering RNA (siRNA) also decreased the phosphorylation of these sites demonstrating that the phosphorylation indeed occurs in a p38-dependent manner (Supplementary Figure 1h). Analysis of amino acid sequence surrounding UV-light-upregulated phosphorylation sites revealed a significant overrepresentation of glutamine (Q) in +1 position, a S/TQ sequence motif that is known to be recognized by ATM/ATR/DNA-PKcs[21,22] (Fig. 1e). To compare the regulatory function of p38 and ATR after UV light, we extracted all UV-light-induced phosphorylation sites that conform to the S/TQ motif. Phosphorylation of 89 S/TQ sites (17%) increased in abundance after UV light (Fig. 1f). The fraction of S/TQ sites within p38-dependent sites was similar to the fraction of S/TQ sites in all quantified phosphorylation sites, indicating that p38-dependent phosphorylation does not target the S/TQ motif (Fig. 1f). In contrast, UV-light-upregulated, p38-dependent phosphorylation occurred within a specific sequence motif that

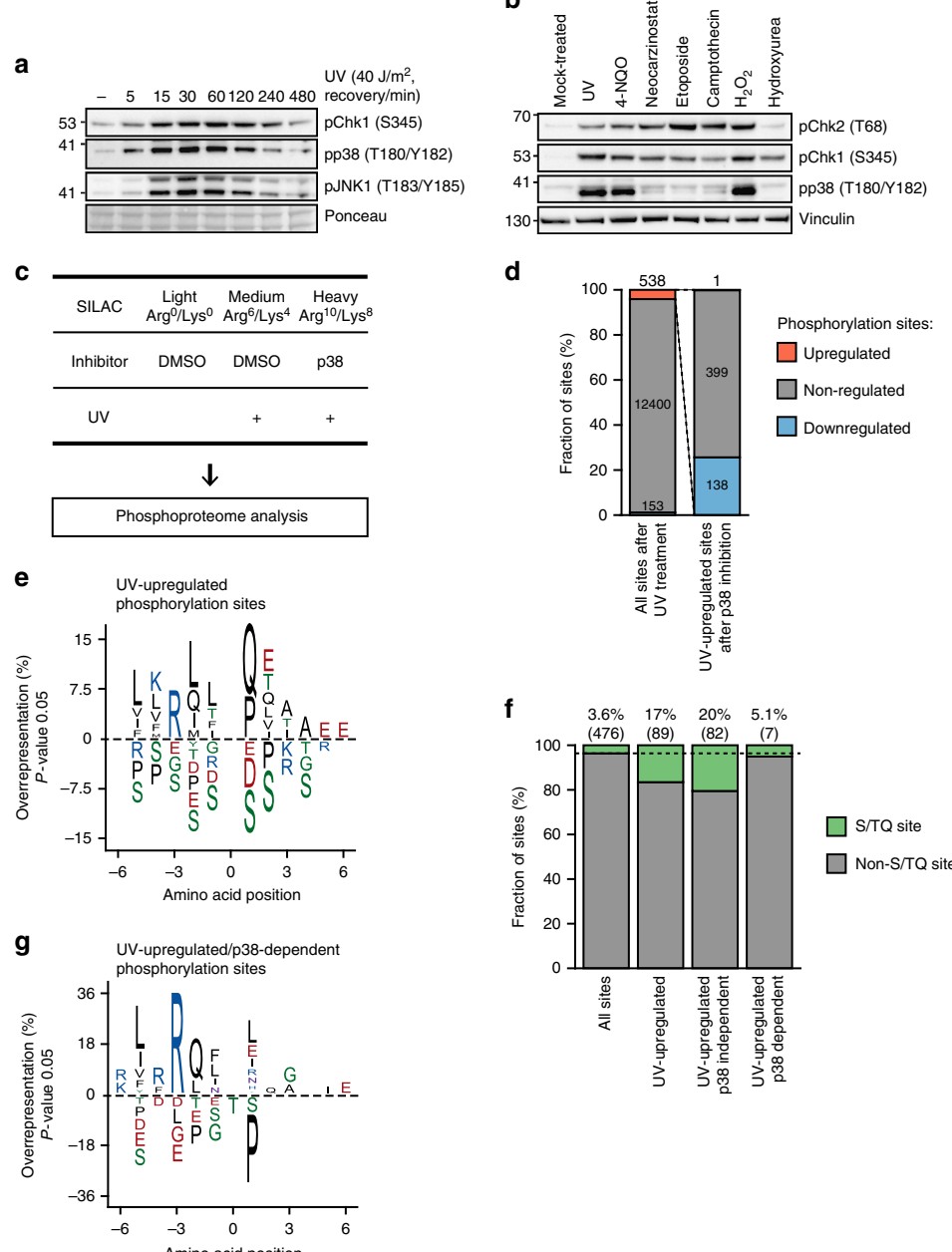

**Fig. 1** Phosphoproteomics reveals a broad scope of p38 signaling after UV light. **a** U2OS cells were treated with UV light (40 J/m$^2$) and left to recover for the indicated times. Total cell lysates were resolved on SDS-PAGE and activation of p38, JNK1, and Chk1 was monitored with phospho-specific antibodies. **b** U2OS cells were left untreated or treated with UV light (40 J/m$^2$, 1 h recovery), 4-nitroquinoline 1-oxide (4-NQO, 20 μM, 1 h), neocarzinostatin (400 μg/ml, 1 h), etoposide (10 μM, 1 h), camptothecin (10 μM, 1 h), H$_2$O$_2$ (2 mM, 1 h), and hydroxyurea (2 mM, 1 h). Total cell lysates were resolved on SDS-PAGE and blotted with the indicated antibodies. **c** Schematic representation of the strategy used to identify UV-light-induced, p38-dependent phosphorylation sites. SILAC-labeled U2OS cells were mock-treated (Light), irradiated with UV light (40 J/m$^2$, 1 h recovery) (Medium), or pretreated with the p38 inhibitor (SB203580, 10 μM, 1 h) or transfected with p38 siRNA, and irradiated with UV light (40 J/m$^2$, 1 h recovery) (Heavy). Phosphorylated peptides were enriched using TiO$_2$ and peptide samples were analyzed by LC-MS/MS. **d** The bar graph shows the number of significantly up-, non-, and downregulated UV-light-induced phosphorylation sites after p38 inhibition identified from two replicate experiments (*p*-value < 0.01, moderated *t*-test). Significantly regulated phosphorylation sites were calculated as shown in Supplementary Figure 1f, g. **e** Sequence motif analysis of 538 UV-light-induced phosphorylation sites. The iceLogo plot shows frequency of six amino acids flanking phosphorylated residue. The frequencies of amino acids surrounding phosphorylated residues in UV-light-induced phosphorylation sites was compared with frequencies in all quantified phosphorylation sites. A significant overrepresentation of phosphorylation sites conforming to the ATM/ATR/DNA-PKcs motif (S/TQ) is observed among UV-light-induced sites. **f** The bar graph shows the absolute number and percentage of S/TQ sites among all quantified phosphorylation sites, UV-light-upregulated sites, UV-light-upregulated, p38-independent sites, and UV-light-upregulated, p38-dependent sites. **g** Sequence motif analysis of 138 UV-light-induced, p38-dependent phosphorylation sites. The analysis was done as described in Fig. 1e

is defined by a glutamine (Q) in position – 2, arginine (R) in – 3, and leucine (L) in – 5 (LXRQXS/T) (Fig. 1g). This motif differs from the p38 motif previously determined using peptide library screening (GPQS/TPI)[23], suggesting that the majority of p38-dependent phosphorylation sites induced after UV light are substrates of kinases acting downstream of p38 rather than p38 itself.

**MK2/3 are key transducers of p38-dependent signaling**. To determine the contribution of the downstream effector kinases to p38-dependent phosphorylation after UV light, we employed SILAC labeling and MS to compare UV light upregulated phosphorylation sites after chemical inhibition of p38 or joint inhibition of MK2, 3, and 5, thereby inhibiting one signaling axis that is activated downstream of p38 (Fig. 2a and Supplementary Figure 2a). Phosphorylation site abundance after p38 and MK2/3/5 inhibition correlated, thus demonstrating that much of the p38-dependent phosphorylation is dependent on the MK2/3/5 signaling axis (Fig. 2b). Importantly, identification of significantly downregulated phosphorylation sites after p38 or MK2/3/5 inhibition showed that nearly 60% of UV-light-upregulated, p38-dependent phosphorylation sites also depend on MK2/3/5 (Fig. 2c and Supplementary Data 2). MK2 and 3 share ~ 65% sequence similarity and previous studies reported that these kinases can

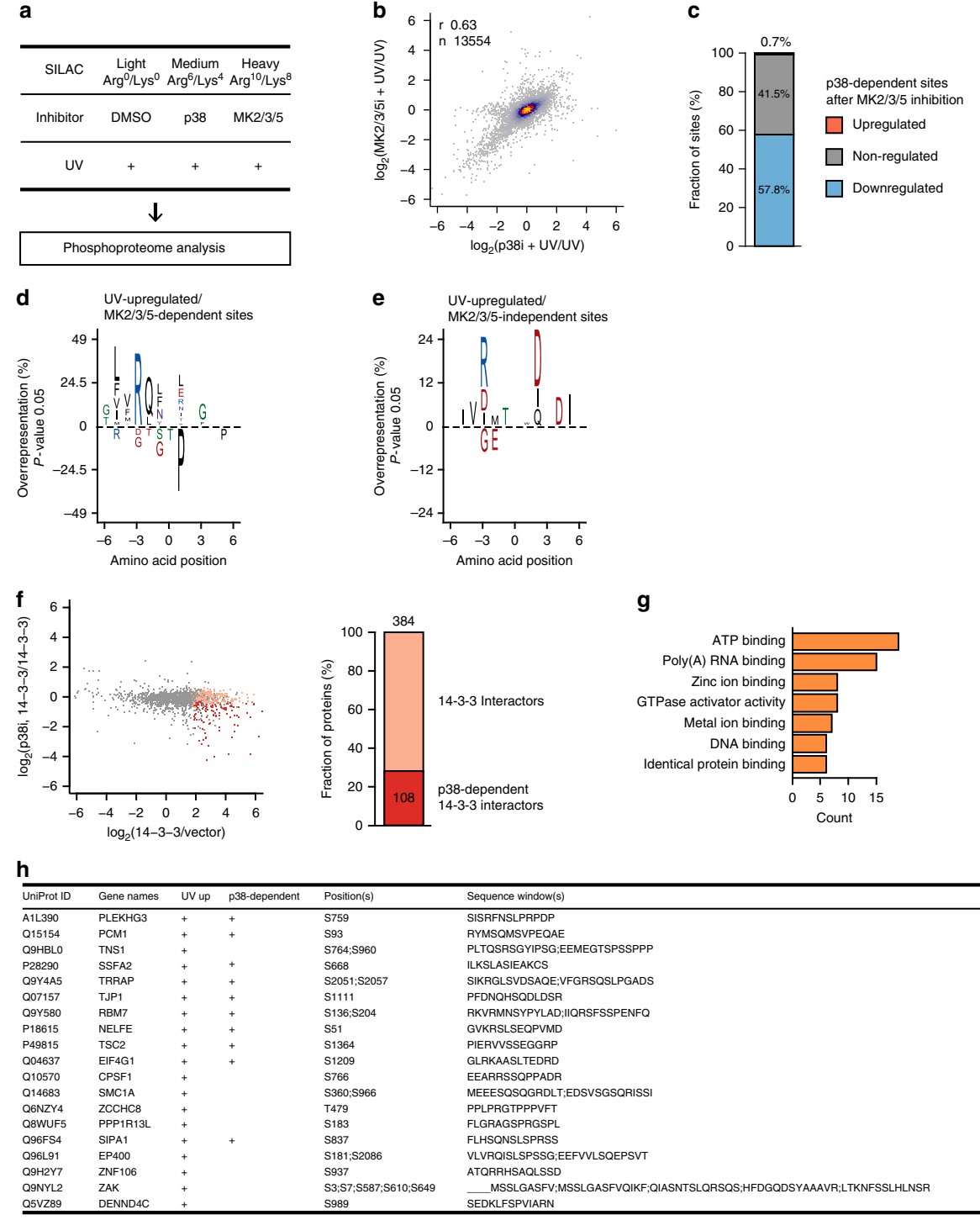

function redundantly, although MK2 tends to show higher activity in cells[24]. To further discriminate the contribution of MK2/3 and MK5 to UV-light-induced phosphorylation, we transiently knocked down MK2/3 or MK5 and monitored phosphorylation of proteins after UV light. These analyses demonstrated that MK2/3 double knockdown decreased the phosphorylation of a majority of MK2/3/5 inhibitor-dependent sites, establishing that MK2 and MK3 are the key transducers of p38 signaling after UV light (Supplementary Data 3). The previously identified p38-dependent sequence motif represents a combination of sequences that are recognized by p38 and its downstream kinases that are activated after UV light (Fig. 1g). Analysis of the sequences surrounding phosphorylation sites that are dependent on both p38 and MK2/3/5 revealed an enrichment of the same motif (LXRQXS/T), demonstrating that this motif is recognized by MK2/3/5 (Fig. 2d). This result is in agreement with a previous study that determined the optimal motif for MK2 in vitro using peptide library screening[23]. Sites phosphorylated independently of MK2/3/5 occurred within an entirely different motif with aspartic acid enriched in position +2 and +4, arginine in −3, and isoleucine in +5 (RXXS/TXDXDI), which likely represents the motif recognized by downstream p38 effectors other than MK2/3/5 that are activated after UV light (Fig. 2e). The motif phosphorylated by MK2/3/5 resembles the motif that can be recognized by proteins of the 14-3-3 family[23,25]. It was previously shown that MK2-dependent phosphorylation of specific substrates can lead to the recruitment of 14-3-3 proteins in response to UV light exposure[26–28]. Therefore, we investigated whether 14-3-3 binding to proteins phosphorylated by p38-MK2 acts as a general regulatory mechanism in cells exposed to UV light. We employed SILAC labeling and MS to compare the interaction profile of 14-3-3 in cells irradiated with UV light and in cells pretreated with the p38 inhibitor and subsequently irradiated with UV light (Supplementary Figure 2b). Notably, this analysis revealed that nearly 30% of identified 14-3-3 interactors bind to 14-3-3 after UV light in a p38-dependent manner (Fig. 2f and Supplementary Data 4). This group of proteins contained a number of RNA-binding proteins, including TNS1, RBM7, NELFE, EIF4G1, CPSF1, SMC1A, and ZNF106 (Fig. 2g, h and Supplementary Data 4).

**p38-MK2/3 signaling axis phosphorylates RNA-binding proteins**. Identification of UV-light-induced p38-dependent phosphorylation sites delivered an unbiased view on the cellular proteins regulated by p38. Notably, Gene Ontology enrichment analysis revealed that proteins with p38-dependent phosphorylation sites are involved in the regulation of messenger RNA stability, gene expression, nuclear-transcribed mRNA poly (A) tail shortening, and translation, and bind to RNA (Fig. 3a). In agreement, p38-dependent phosphorylation sites occurred on proteins in the nucleolus, cytoplasmic stress granules, and focal adhesions, showing that p38 regulates proteins in the nucleus and cytoplasm (Fig. 3a). RNA binding was also the most significantly enriched term among all proteins containing UV-light-upregulated sites, demonstrating that UV-light-induced phosphorylation of RNA-binding proteins, which is predominantly executed by the p38-MK2/3 signaling axis, is a hallmark of the cellular response to UV light (Supplementary Figure 3a). Proteins containing UV-light-induced S/TQ phosphorylation sites that are likely direct substrates of ATR are involved in DNA repair and cell cycle, and primarily localized in the nucleus (Fig. 3b). MLH1, PMS2, NBN, FANCI, TOPBP1, RAD1, and Chk1 display UV-light-upregulated phosphorylation on S/TQ sites and are engaged in functional networks involved in DNA damage repair and signaling (Supplementary Figure 3b). In addition, proteins previously not linked to DNA repair also contain UV-light-induced S/TQ phosphorylation, suggestive of their function in the DNA damage response (Supplementary Figure 3b). In contrast, proteins with p38-dependent phosphorylation sites are engaged in functional networks involved in the regulation of RNA metabolism, but not DNA repair (Fig. 3c). Among the proteins with p38-dependent phosphorylation sites were subunits of the NELF (NELFE and NELFA) and SMN complex (GEMIN5 and DDX20), proteins involved in the degradation of ARE-containing mRNA (PARN, KHSRP, and ZFP36), mRNA polyadenylation (WDR33 and SYMPK), and translation (EIF5, EIF4G1, SRP54, and DHX29) (Fig. 3c).

**NELFE transiently interacts with 14-3-3 after UV light**. We identified the RNA-binding subunit of the NELF complex NELFE as a substrate of p38-MK2-dependent phosphorylation after UV light. The NELF complex inhibits transcriptional elongation of RNA pol II in *Drosophila* and mammalian cells. Productive RNA pol II elongation into the gene body is achieved by phosphorylation of the C-terminal domain of RNA pol II on serine 2, as well as NELFE by P-TEFb complex containing cyclin T and CDK9[29,30]. Affinity purification of NELFE showed that UV light induces the binding of NELFE to different 14-3-3 proteins (Fig. 4a). To examine whether UV-light-induced interaction between NELFE and 14-3-3 is dependent on NELFE phosphorylation, we performed pull downs from cells treated with the p38 or MK2/3/5 inhibitor before irradiation with UV light. Notably,

**Fig. 2** MK2/3 are key transducers of p38 signaling after UV light. **a** Schematic representation of the strategy used to identify UV-light-induced, MK2/3/5-dependent phosphorylation sites. SILAC-labeled U2OS cells were mock-treated (Light), pretreated with the p38 inhibitor (Medium) or the MK2/3/5 inhibitor (Heavy), and subsequently irradiated with UV light (40 J/m$^2$, 1 h recovery). The phosphoproteome analysis was performed as described in Fig. 1c. **b** The scatter plot shows the logarithmized SILAC ratios of quantified phosphorylation sites. The color-coding indicates the density. A majority of UV-light-induced, p38-dependent phosphorylation sites significantly decreased in abundance also after MK2/3/5 inhibition, whereas a smaller fraction of sites decreased in abundance only after p38 inhibition. **c** The bar graph shows the percentage of UV-light-upregulated, p38-dependent sites that are up-, non-, or downregulated after LXRQXS/5 inhibition. Nearly 60% of p38-dependent phosphorylation sites are also dependent on MK2/3/5. **d** Sequence motif analysis of p38- and MK2/3/5-dependent phosphorylation sites. The analysis was done as described in Fig. 1e. The sequence surrounding the phosphorylated residue shows an enrichment of glutamine (Q) in position −2, arginine (R) in −3, and leucine (L) in −5. **e** Sequence motif analysis of p38-dependent, MK2/3/5-independent sites phosphorylation sites. The analysis was done as described in Fig. 1e. The sequence surrounding the phosphorylated residue shows an enrichment of aspartic acid (D) in position +2, +4, and arginine (R) in −3. **f** The scatter plot shows the logarithmized SILAC ratios of quantified proteins. 14-3-3 interaction partners and p38-dependent interactions are indicated in light and dark red, respectively. Three hundred and eighty-four proteins were significantly enriched in 14-3-3 pull downs after UV light (p-value < 0.05, moderated t-test). One hundred and eight out of 384 proteins (28%) bound to 14-3-3 in a p38-dependent manner. **g** The bar plot shows the number of proteins with the indicated Gene Ontology (GO)-molecular function terms that were enriched among p38-dependent 14-3-3 interaction partners. **h** The table shows selected RNA-binding proteins that were identified as p38-dependent interaction partners of 14-3-3. The UniProt ID, protein name, gene name, position(s), and sequence window of UV-light-induced phosphorylation sites identified in phosphoproteomics screen is indicated

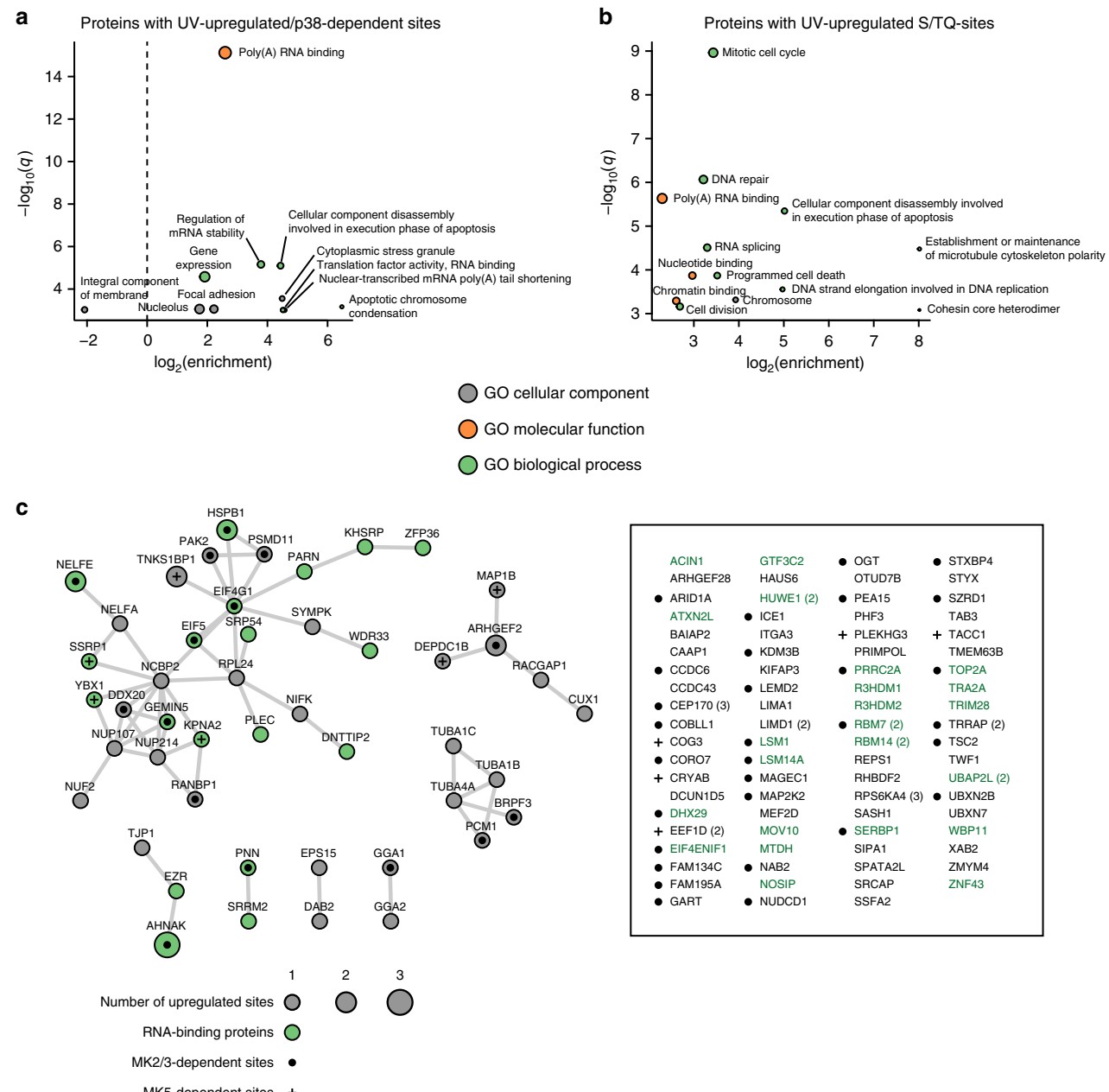

**Fig. 3** p38 phosphorylates RNA-binding proteins after UV light. **a** GO terms significantly enriched among proteins with UV-light-upregulated, p38-dependent phosphorylation sites. The dot plot shows significantly enriched GO terms associated with proteins containing p38-dependent phosphorylation sites compared with proteins with non-regulated phosphorylation sites. The significance of the enrichment of a specific term was determined using Fisher's exact test. P-values were corrected for multiple hypotheses testing using the Benjamini and Hochberg FDR. **b** GO terms significantly enriched among proteins with UV-light-upregulated, S/TQ phosphorylation sites. The dot plot shows significantly overrepresented GO terms associated with proteins containing S/TQ phosphorylation sites compared with proteins with non-regulated phosphorylation sites. The analysis was done as described in Fig. 3a. **c** Analysis of functional interactions among proteins with UV-light-upregulated, p38-dependent phosphorylation sites. The functional interactions were obtained from the STRING database and visualized using Cytoscape. Proteins with UV-light-upregulated, p38-dependent sites that do not form a functional network are indicated on the right and proteins annotated with "RNA-binding" GO-molecular function term are indicated in green. Proteins with MK2/3-dependent phosphorylation sites are indicated with circles and proteins with MK5-dependent sites with crosses

the binding between NELFE and 14-3-3 was completely abolished after p38 or MK2/3/5 inhibition (Fig. 4b). Knockdown of p38 or MK2 also inhibited the binding between NELFE and 14-3-3, demonstrating that MK2 and not MK3 or MK5 is responsible for the phosphorylation of NELFE in U2OS cells (Fig. 4c). The interaction between NELFE and 14-3-3 after UV light and its dependency on p38 was also validated in HaCaT, HEK293T, and RPE-1 cells, demonstrating that this interaction is not restricted

to U2OS cells (Supplementary Figure 4a). UV-light-induced, p38-dependent phosphorylation of NELFE could also be readily detected by western blotting using phospho-specific antibodies recognizing the 14-3-3-binding motif (Fig. 4d). To investigate whether P-TEFb, which phosphorylates the NELF complex in unperturbed cells, has a role in regulating the interaction between NELFE and 14-3-3 after cellular stress, we inhibited its activity using 5,6-dichloro-1-β-D-ribofuranosylbenzimidazole (DRB).

Inhibition of P-TEFb did not inhibit but rather augmented NELFE interaction with 14-3-3 (Fig. 4e). NELFE interaction with 14-3-3 occurred already 15 min after UV light exposure and peaked 30–60 min post irradiation, indicating that NELFE binding to 14-3-3 is dynamic and has a role early after exposure of cells to UV light (Fig. 4f). The interaction of NELFE with 14-3-3 also occurs after oxidative stress that leads to rapid activation of p38 (Fig. 1b and Supplementary Figure 4b). It was recently shown that XPC knockdown decreases the activation of p38 after UV light[31]. Notably, we found that knockdown of CSB or XPC resulted in decreased binding of 14-3-3 to NELFE, which is in

agreement with the reduced activation of p38 in these cells and suggests that the interaction is partially dependent on the DNA damage recognition by the NER machinery (Fig. 4g).

**NELFE phosphorylation by MK2 regulates its binding to 14-3-3.** We found that NELFE was phosphorylated on eight serine residues after UV light (Supplementary Data 1 and 2). Notably, UV-light-induced phosphorylation of NELFE on serine 49, 51, 115, and 251 was dependent on p38 and MK2 (Fig. 5a). In accordance with our data that NELFE binds to 14-3-3 after UV

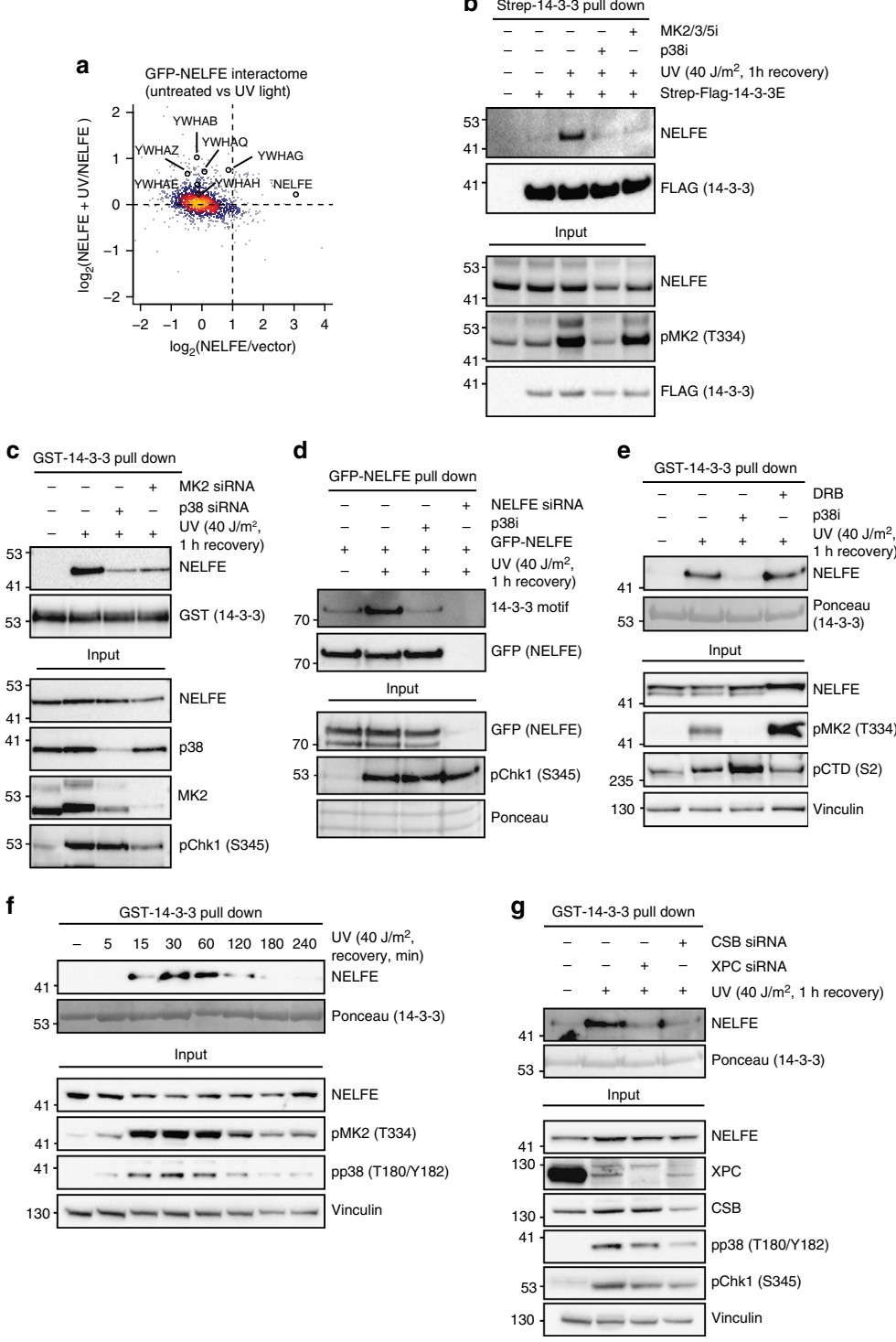

light, computational analysis of the sequence surrounding these phosphorylation sites predicted that serine 51, 115, and 251 reside within 14-3-3-binding motifs (Fig. 5b)[32]. To determine the importance of the identified phosphorylation sites for UV-light-induced binding of NELFE to 14-3-3, we mutated serine 49/51, 115, and 251 to alanine and performed pull downs using GST-14-3-3 and lysates expressing NELFE serine-to-alanine mutants. Wild-type green fluorescent protein (GFP)-NELFE was efficiently pulled down using GST-14-3-3, whereas the S115A mutant of NELFE did not interact with GST-14-3-3 (Fig. 5c). NELFE S49/51A and S251A bound weaker to GST-14-3-3, indicating that these phosphorylation sites also contribute to the interaction (Fig. 5c). Serine 115 phosphorylation of NELFE increased twofold and phosphorylation site occupancy analysis revealed that this phosphorylation affects nearly the complete cellular pool of NELFE after UV light, indicative of the physiological importance of this modification (Fig. 5d,e and Supplementary Figure 5a). The absolute occupancy of phosphoserine 49, 51, and 251 was lower than phosphoserine 115, but substantially increased after UV light, in line with the results that these phosphorylation sites also contribute to the binding between NELFE and 14-3-3 (Fig. 5e). Recombinant MK2 could also phosphorylate NELFE on serine 51, 115, and 251 in vitro (Supplementary Figure 5b). To confirm that serine 115 phosphorylation is required for binding of NELFE to 14-3-3, we compared the interaction partners of GFP-tagged wild-type NELFE and S115A mutant after UV light. Indeed, this experiment showed that endogenous 14-3-3 does not bind to NELFE S115A mutant (Fig. 5f). Interestingly, serine 115 in NELFE is highly conserved in evolution, suggesting that phosphorylation of this residue is of regulatory importance also in other organisms (Supplementary Figure 5c). To investigate whether the binding between NELFE and 14-3-3 after UV light is direct, we synthesized a biotinylated phospho-peptide centered around serine 115 and performed pull downs with recombinant 14-3-3. We could observe binding of 14-3-3 to the phosphorylated peptide, whereas no binding was detected if the peptide was dephosphorylated before the pull down (Fig. 5g).

To further investigate the binding mode of NELFE and 14-3-3, we determined the crystal structure by molecular replacement using the previously reported 14-3-3 epsilon structure as a search model (PDB: 2BR9). The crystal structure of 14-3-3 epsilon in complex with the phosphorylated NELFE peptide revealed that the overall structure of the nine helices (αA to αI) of 14-3-3 and the peptide orientation were similar as previously reported for complexes of 14-3-3 epsilon and other phospho-peptides[33] (Fig. 5h). It was shown that 14-3-3 proteins form hetero and/or homo dimers, and that 14-3-3 epsilon preferentially forms hetero

dimers[34]. Although the asymmetric unit of the crystal contained one 14-3-3 epsilon and one phosphorylated peptide (Fig. 5h, Yellow), 14-3-3 epsilon forms homo dimers with symmetry-related neighboring molecules in the crystal (Fig. 5h and Supplementary Figure 5d, Cyan). We observed a decrease in binding between NELFE and 14-3-3 upon mutation of serine 49/51 and serine 251 to alanine, which suggests that one of these phosphorylation sites serves as a secondary binding surface for the 14-3-3 homo or hetero dimer in vivo. Interestingly, we also found that the NELFA subunit of the NELF complex contains a UV-light-induced, p38-dependent phosphorylation site that is predicted to bind 14-3-3 and could thus provide an alternative secondary binding surface for the 14-3-3 dimer. As reported for other 14-3-3-phospho-peptide complexes, the N-terminal four helices, αA, αB, αC, and αD, are essential and two salt bridges between Arg19 on one molecule and Glu92 on another are the driving force for the 14-3-3 dimer formation[35]. In addition, Tyr85 sitting under the conserved lysine position on αD forms an accessory hydrogen bond with Glu22, supporting homo dimer formation in the crystal (Supplementary Figure 5d,e). 14-3-3 epsilon has a peptide-binding groove, composed of αC, αE, αG, and αI that catches the phosphorylated peptide. Especially the conserved triad of Arg57, Arg130, and Tyr131 produces a positively charged patch, directly interacting with the phosphate group of the phosphorylated peptide (Fig. 5h). In addition, a few hydrogen bonds and salt bridges between the main chain of the phosphorylated peptide and 14-3-3 epsilon contribute to the complex formation and adjust its relative orientation (Fig. 5h).

**p38 promotes dissociation of the NELF complex from chromatin.** To study the functional consequence of p38-dependent phosphorylation of NELFE and its possible impact on transcription after UV light, we analyzed whether the composition of the chromatin proteome is altered upon inhibition of p38. To this end, we isolated chromatin-associated proteins from cells and used SILAC-based quantitative MS to monitor the p38-dependent changes in the chromatin proteome after UV light (Supplementary Figure 6a,b). We identified 48 proteins that were recruited to and 44 that dissociated from chromatin after exposure of cells to UV light (p-value < 0.01, moderated t-test, Supplementary Data 5). In addition to DNA repair factors, which comprised a large group of proteins that were enriched on chromatin, we also found a significant enrichment of 24 proteins annotated with the Gene Ontology (GO) term "RNA-binding" on chromatin (Supplementary Data 5). In particular, this group included 14 proteins functioning in the ribosome biogenesis and annotated with the

---

**Fig. 4** UV-light-induced phosphorylation of NELFE by MK2 leads to 14-3-3 binding. **a** Identification of p38-dependent NELFE interaction partners after UV light. SILAC-labeled U2OS cells expressing GFP-NELFE were mock-treated or irradiated with UV light. Cells were lysed and protein extracts were incubated with GFP Trap agarose. Enriched proteins were resolved on SDS-PAGE and digested in-gel into peptides. Peptides were extracted from gel and analyzed by LC-MS/MS. The scatter plot shows the logarithmized SILAC ratios of proteins quantified in the pull down. The color coding indicates the density. **b** NELFE interaction with 14-3-3 after UV light is p38- and MK2/3/5-dependent. U2OS cells expressing Flag-Strep-14-3-3 or an empty vector were mock-treated, irradiated with UV light or pretreated with the p38 or MK2/3/5 inhibitor, and then irradiated with UV light. Cells were lysed and protein extracts were incubated with StrepTactin sepharose. Enriched proteins were resolved by SDS-PAGE and selected proteins were detected with the indicated antibodies. **c** NELFE interaction with GST-14-3-3 is abolished in p38 and MK2 knockdown cells. U2OS cells were transfected with non-targeting, p38, or MK2-targeting siRNA and then irradiated with UV light. Cells were lysed and protein extracts were incubated with recombinant GST-14-3-3. Enriched proteins were resolved by SDS-PAGE and NELFE was detected using a specific antibody. **d** NELFE is phosphorylated after UV light on a 14-3-3-binding motif. GFP-NELFE was pulled down using GFP Trap agarose. Phosphorylation of NELFE was detected using antibodies recognizing the 14-3-3 motif. NELFE knockdown was used as control. **e** NELFE interacts with 14-3-3 after inhibition of P-TEFb. U2OS cells were treated with the p38 inhibitor or P-TEFb inhibitor 5,6-dichloro-1-β-D-ribofuranosylbenzimidazole (DRB) and then irradiated with UV light. After cell lysis, protein extracts were incubated with the recombinant GST-14-3-3. **f** Dynamics of NELFE interaction with 14-3-3 after UV light. U2OS cells were exposed to UV light and left to recover for the indicated time points. After cell lysis, protein extracts were incubated with the recombinant GST-14-3-3. **g** NELFE interaction with 14-3-3 is partially dependent on the NER machinery. U2OS cells were transfected with a non-targeting siRNA or siRNA targeting XPC or CSB and then irradiated with UV light. After cell lysis, protein extracts were incubated with the recombinant GST-14-3-3

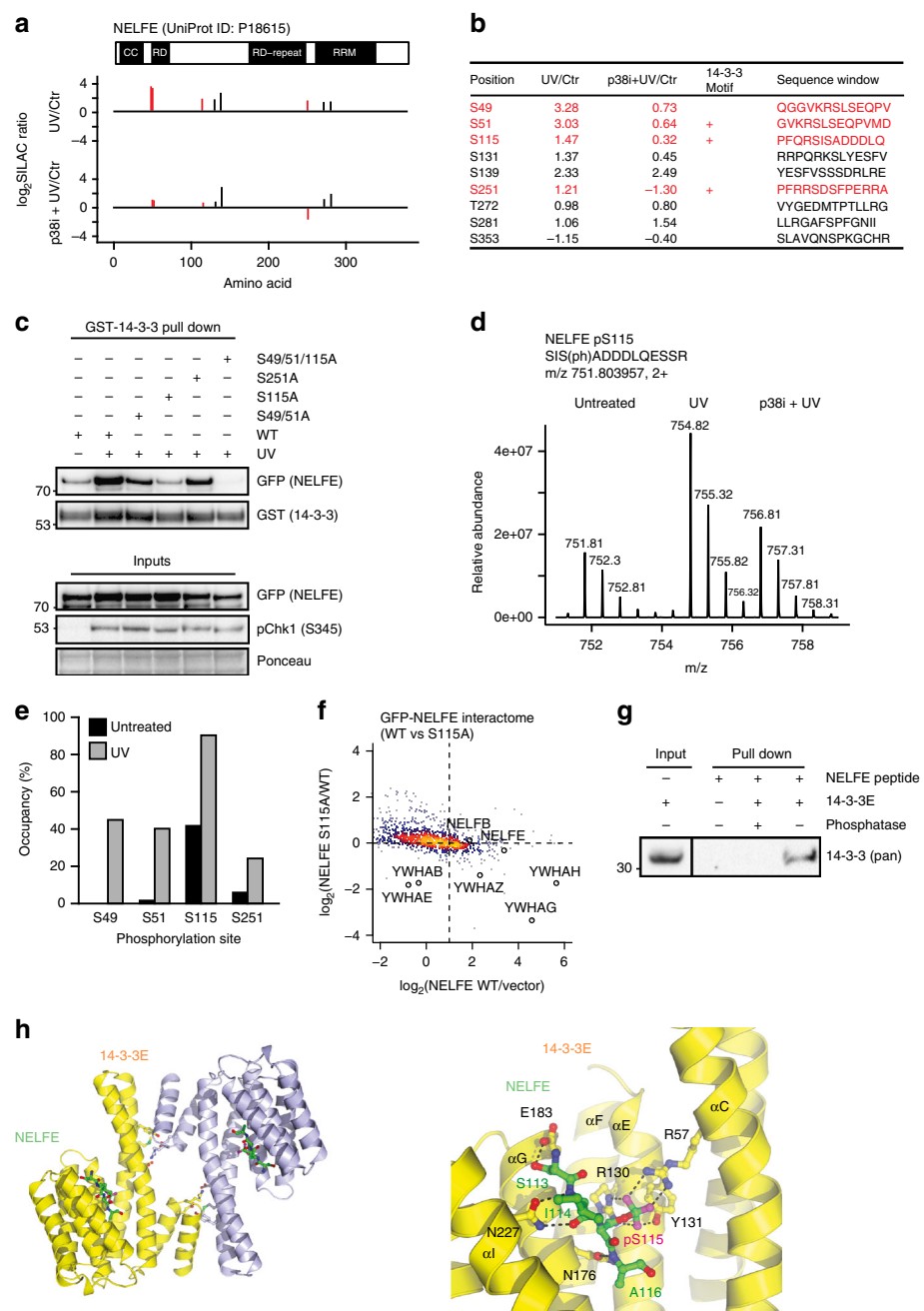

**Fig. 5** NELFE phosphorylation on S115 is required for the interaction with 14-3-3. **a** Schematic representation of NELFE domain organization and phosphorylation sites that were identified by phosphoproteomics. The SILAC ratios quantified for phosphorylation sites on NELFE after UV light and p38 inhibition are indicated. UV-light-induced, p38-dependent phosphorylation sites are labeled in red. **b** The table shows all phosphorylation sites identified on NELFE by phosphoproteomics. The position, SILAC ratios, 14-3-3 binding prediction and sequence window are indicated. UV-light-induced, p38-dependent phosphorylation sites are labeled in red. **c** Serine 115 phosphorylation is required for the interaction of NELFE and 14-3-3. U2OS cells expressing GFP-tagged wild-type NELFE or NELFE serine-to-alanine mutants were irradiated with UV light. Protein extracts were incubated with GST-14-3-3 and enriched proteins were resolved on SDS-PAGE. **d** Mass spectrometric parent ion scan of the peptide SISADDDLQESSR corresponding to S115 in NELFE. The SILAC triplet shows the relative abundance and mass to charge (m/z) of the phosphorylated peptide in mock-treated cells and cells irradiated with UV light without or with pretreatment with the p38 inhibitor. **e** Absolute occupancy of serine 49, 51, 115, and 251 phosphorylation in NELFE in undamaged cells and after UV light was determined by MS. **f** NELFE S115A mutant does not bind to 14-3-3. SILAC-labeled cells overexpressing GFP-tagged wild-type NELFE or NELFE S115A mutant were irradiated with UV light. UV-light-irradiated U2OS cells overexpressing GFP alone were used as control. Cells were lysed and protein extracts were incubated with GFP Trap agarose. The scatter plot shows the logarithmized SILAC ratios of quantified proteins. The color-coding indicates the density. **g** Recombinant 14-3-3 binds to phosphorylated NELFE peptide. Biotinylated phosphorylated NELFE peptide corresponding to serine 115 was bound to NeutrAvidin agarose. Phosphorylated and dephosphorylated peptide were incubated with purified 14-3-3. **h** Structure of 14-3-3 epsilon in complex with NELFE phosphorylated peptide QPFQRSI(p)SADDDLQE. Structure of the 14-3-3 epsilon in cartoon representation (Yellow and Cyan) and NELFE phosphorylated peptide in ball and stick model (Green). The inset on the right shows the 14-3-3 epsilon–NELFE phosphorylated peptide interaction

GO term "maturation of SSU-rRNA from tricistronic rRNA transcript (SSU-rRNA, 5.8 S rRNA, LSU-rRNA)" (Fig. 6a). Well-known DNA repair factors were recruited to chromatin after exposure of cells with UV light, including RPA1/2, PCNA, XPA, MSH2/6, PMS1, MLH1, and FANCD2 (Fig. 6b). Some DNA repair factors including Chk1, DDB1/2, and CCAR2 were excluded from chromatin in response to UV light; however, the recruitment or removal of DNA repair factors was not affected by p38 inhibition (Fig. 6b). UV light resulted in the rapid dissociation of some RNA-binding proteins from chromatin including all components of the NELF complex (Fig. 6c). As previously reported, we also observed UV light and p38-dependent removal of the NEXT complex subunits RBM7 and ZCCH8 from chromatin[26,36]. Notably, inhibition of p38 abolished UV-light-induced dissociation of NELF complex from chromatin (Fig. 6c, d and Supplementary Figure 6c). UV light exposure resulted in NELFE dissociation from chromatin that was dependent on p38 activity also in the human keratinocyte cell line (HaCaT) (Supplementary Figure 6d). Monitoring the levels of NELFE on chromatin at different time points post UV light irradiation revealed that NELFE levels on chromatin returned to normal as cells recovered from DNA damage 48 h after irradiation (Fig. 6e and Supplementary Figure 6e). In line with this, we found that inhibiting NELFE recovery on chromatin by transient knock-down resulted in the increased sensitivity of cells to UV-light-induced DNA damage (Fig. 6f).

**NELFE release is accompanied by transcriptional elongation**. The NELF complex interacts with RNA pol II at promoter-proximal sites to inhibit transcriptional elongation[15]. To further study whether NELF complex dissociation from chromatin after UV light correlates with changes in RNA pol II chromatin-binding genome-wide, we performed chromatin immunoprecipitation sequencing (ChIP-seq) of RNA pol II in untreated cells and after UV light exposure (40 J/m$^2$, 1 h recovery). As expected, in untreated cells we could detect a clear enrichment of RNA pol II around transcription start sites (TSSs) (Fig. 7a). Exposure of cells to UV light resulted in a slight decrease in RNA pol II enrichment at TSSs that can occur as consequence of RNA pol II degradation, inhibition of transcriptional initiation, or enhanced RNA pol II release into downstream regions caused by the NELF complex dissociation from chromatin (Fig. 7a). In line with our results, a recent study that analyzed transcription in cells exposed to UV light reported a reduction of nascent transcripts in promoter-proximal regions[12]. We first tested whether NELF complex dissociation from chromatin leads to degradation of RNA pol II by quantifying the levels of different RNA pol II subunits on chromatin by SILAC-based MS. UV light did not result in a decreased level of RNA pol II on chromatin 1 h post-irradiation (Supplementary Figure 7a). On the contrary, the levels of RNA pol II on chromatin slightly increased at this time point (Supplementary Figure 7a). To test whether UV light leads to RNA pol II release into downstream regions of genes, we calculated the RNA pol II release ratio (polymerase release ratio, PRR) that is defined as a ratio of the RNA pol II signal intensity in downstream regions of genes to the signal intensity at promoters (Supplementary Figure 7b)[37]. From the RNA pol II ChIP-seq, we calculated PRRs for 6,898 RNA pol II target genes in U2OS cells. These analyses revealed a significant increase in the PRR of 2,123 genes after UV light at 1 h time point post irradiation compared with mock-treated cells (Fig. 7b and Supplementary Data 6). In contrast, only 25 genes showed a decrease of the PRR when applying the same significance threshold (Supplementary Data 6). In support of these results, reanalysis of the nascent RNA-sequencing (RNA-seq) from Williamson et al.[12] also revealed a general increase in PRRs after UV light irradiation (Supplementary Figure 7c). Gene-set enrichment analysis revealed that genes with upregulated PRRs after UV light are involved in telomere maintenance, RNA metabolism, cell cycle, DNA repair, and RAS/ERK signaling (Fig. 7c, d and Supplementary Figure 7d). Notably, a comparison of these genes with NELFE targets determined by ChIP-seq in HeLa cells[38] identified that 70% of genes that displayed an increase in the PRR after UV light are also bound by NELFE (Supplementary Figure 7e). Taken together, our results demonstrate that UV light exposure of human cells results in increased RNA pol II elongation in a subset of genes, which temporally correlates with the p38-MK2-dependent NELF complex release from chromatin (Fig. 7e and Supplementary Figure 7f).

## Discussion

Exposure of human cells to UV light induces the formation of bulky UV photo-products that interfere with DNA replication and transcription[1]. To maintain genome stability, cells need to coordinate DNA repair with cell cycle progression, DNA replication, and RNA metabolism.

Protein phosphorylation dependent on ATR-Chk1 plays an integral role in DNA repair and cell cycle checkpoint activation after UV light[2]; however, the function of other kinase-dependent signaling pathways remains poorly understood. In this study, we demonstrate that UV light triggers widespread and rapid phosphorylation of RNA-binding proteins that is dependent on the p38 MAPK pathway and directly mediated by the p38 effector kinase MK2. We identify 138 sites present on 122 proteins that are phosphorylated in a p38-MK2-dependent manner. Moreover, we show that many of these phosphorylation sites serve as platform for the recruitment of 14-3-3 proteins. Previous studies reported that phosphorylation of specific proteins by MK2 after UV light can lead to binding of 14-3-3 and thereby regulate cell cycle, turnover of non-coding RNA, and remodeling of centriolar satellites[26–28]. We establish now that p38-MK2-dependent recruitment of 14-3-3 to RNA-binding proteins provides a broad regulatory mechanism functioning rapidly upon exposure of cells to UV light.

It was previously shown that UV light globally affects different levels of RNA metabolism, including transcription, RNA splicing, and translation[6]. However, the signaling pathways and mechanisms that regulate RNA metabolism after UV light remain poorly understood. Recent studies that employed nascent RNA-seq to monitor transcription after UV light reported changes in transcriptional elongation[12,18,19]. Many of the identified p38-MK2 protein substrates are functioning in the regulation of transcription: we show that the NELF complex is a substrate of p38-MK2-dependent phosphorylation after exposure of cells to UV light. In human cells, the NELF complex (comprising the four subunits NELFA, NELFB, NELFCD, and NELFE) inhibits RNA pol II elongation shortly after initiation to induce promoter-proximal pausing[15]. Promoter-proximal pausing was suggested to occur in many, if not all, genes; however, it seems to have a particularly important role in the regulation of developmental and stimuli-induced genes[39–44]. Phosphorylation of the RNA-binding subunit NELFE on S115 by MK2 promotes its binding to 14-3-3 and dissociation of the NELF complex from chromatin that is accompanied by RNA pol II elongation in genes functioning in telomere maintenance, RNA metabolism, cell cycle, and DNA repair. Knockdown of NELFE subunit of the NELF complex was shown to result in global RNA pol II elongation in unstressed primary cells and in cancer cells[45,46]. In agreement with our results, a recent study reported that UV light triggers RNA pol II elongation and transcription of active genes, which leads to

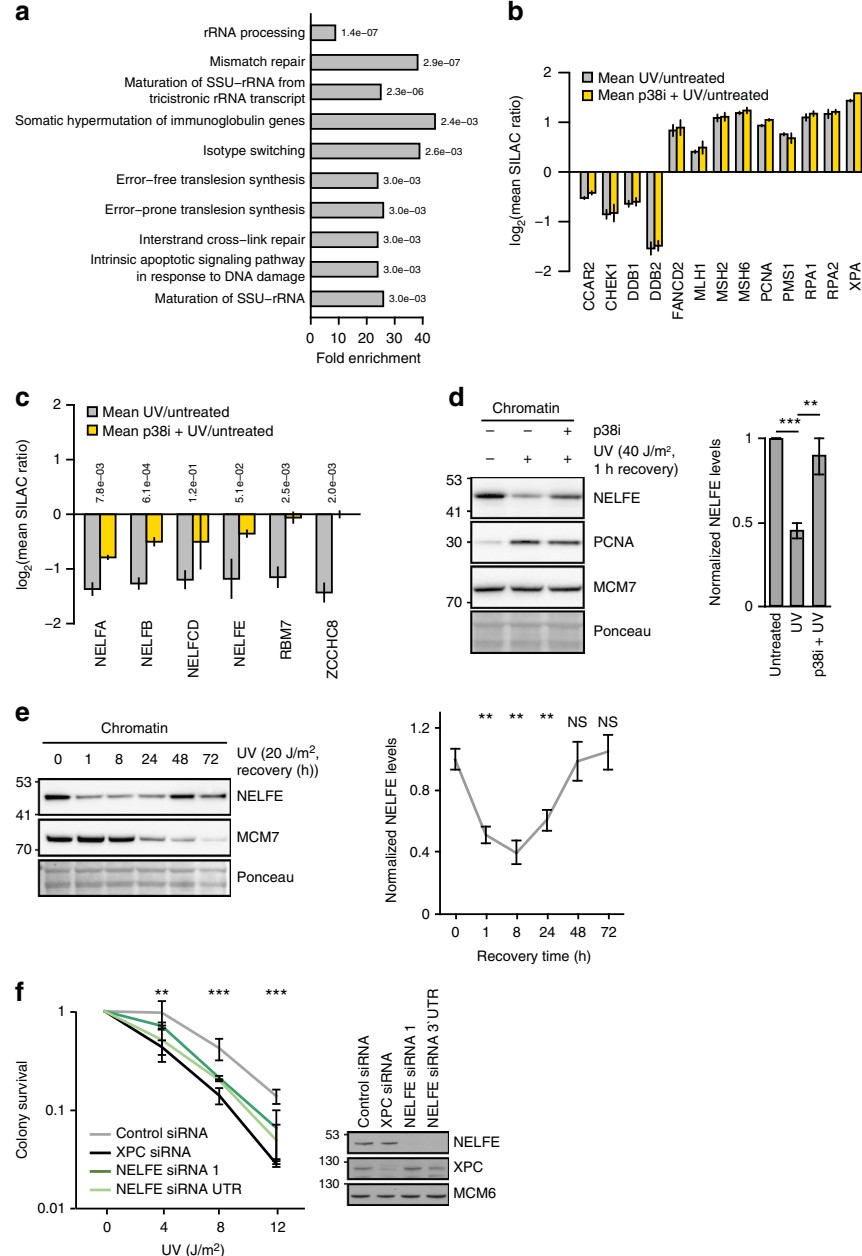

**Fig. 6** p38-dependent phosphorylation of NELFE promotes its dissociation from chromatin. **a** Chromatin-associated proteins were extracted from untreated, UV-light-treated, and p38i, UV-light-treated U2OS cells, and analyzed by SILAC-based quantitative mass spectrometry. The bar plot shows the GO-BP (biological process) terms associated with proteins specifically enriched on or removed from chromatin after UV light (40 J/m², 1 h recovery). The significance of the enrichment of a specific term was determined using Fisher's exact test. *P*-values were corrected for multiple hypotheses testing using the Benjamini and Hochberg FDR. **b** The bar plot shows selected proteins associated with DNA repair and cell cycle that are significantly recruited or removed from chromatin after UV light. The error bars show the mean and SD of SILAC ratios quantified from three replicate experiments. Two-sided Student's *t*-test was used to assess the significance. **c** The NELF complex subunits are removed from chromatin in a UV light and p38-dependent manner. The bar plot shows selected proteins whose removal from chromatin after UV light is dependent on p38. The error bars show the mean and SD of SILAC ratios quantified from three replicate experiments. Two-sided Student's *t*-test was used to assess the significance. **d** NELFE dissociates from chromatin after UV light. Chromatin protein fractions from differentially treated U2OS cells were resolved by SDS-PAGE and subjected to western blotting with the indicated antibodies (left). The levels of NELFE on chromatin were quantified from three replicate experiments and normalized to MCM7 levels (right). The error bars show the mean and SD of SILAC ratios quantified from three replicate experiments. Two-sided Student's *t*-test was used to assess the significance (***p*-value < 0.001, ***p*-value < 0.01). **e** Dynamics of NELFE removal from chromatin. Chromatin protein fractions from differentially treated U2OS cells were resolved by SDS-PAGE and subjected to western blotting with the indicated antibodies. The error bars show the mean and SD of SILAC ratios quantified from three replicate experiments. Two-sided Student's *t*-test was used to assess the significance (***p*-value < 0.01). **f** Knockdown of NELFE reduced the ability of U2OS cells to form colonies after UV light. The error bars show the mean and SD of results obtained in three replicate experiments performed in three technical replicates. Two-sided Student's *t*-test was used to assess the significance (***p*-value < 0.001, ***p*-value < 0.01)

enhanced DNA damage sensing by RNA pol II and DNA repair through the TC-NER pathway[19]. In addition to the NELF complex, we found that other proteins involved in transcriptional elongation, including TRIM28 and LARP7[47,48], are phosphorylated by p38, suggesting that multiple p38-MK2-mediated phosphorylation-dependent events cooperate to regulate RNA pol II elongation after UV light. Moreover, NELFE ADP-ribosylation has been shown to promote RNA pol II elongation in unstressed cells[45]. It is possible that different posttranslational

modifications of NELFE, including phosphorylation and ADP-ribosylation, regulate RNA pol II elongation in response to UV light. Dissociation of the NELF complex from chromatin by a different mechanism mediated by enhancer RNAs was shown to promote the induction of immediate early genes in response to an increase in neuronal activity[40].

Our study links NELF complex regulation with the cellular response to UV-light-induced DNA damage. The NELF complex was recently associated with the double-strand break (DSB) repair

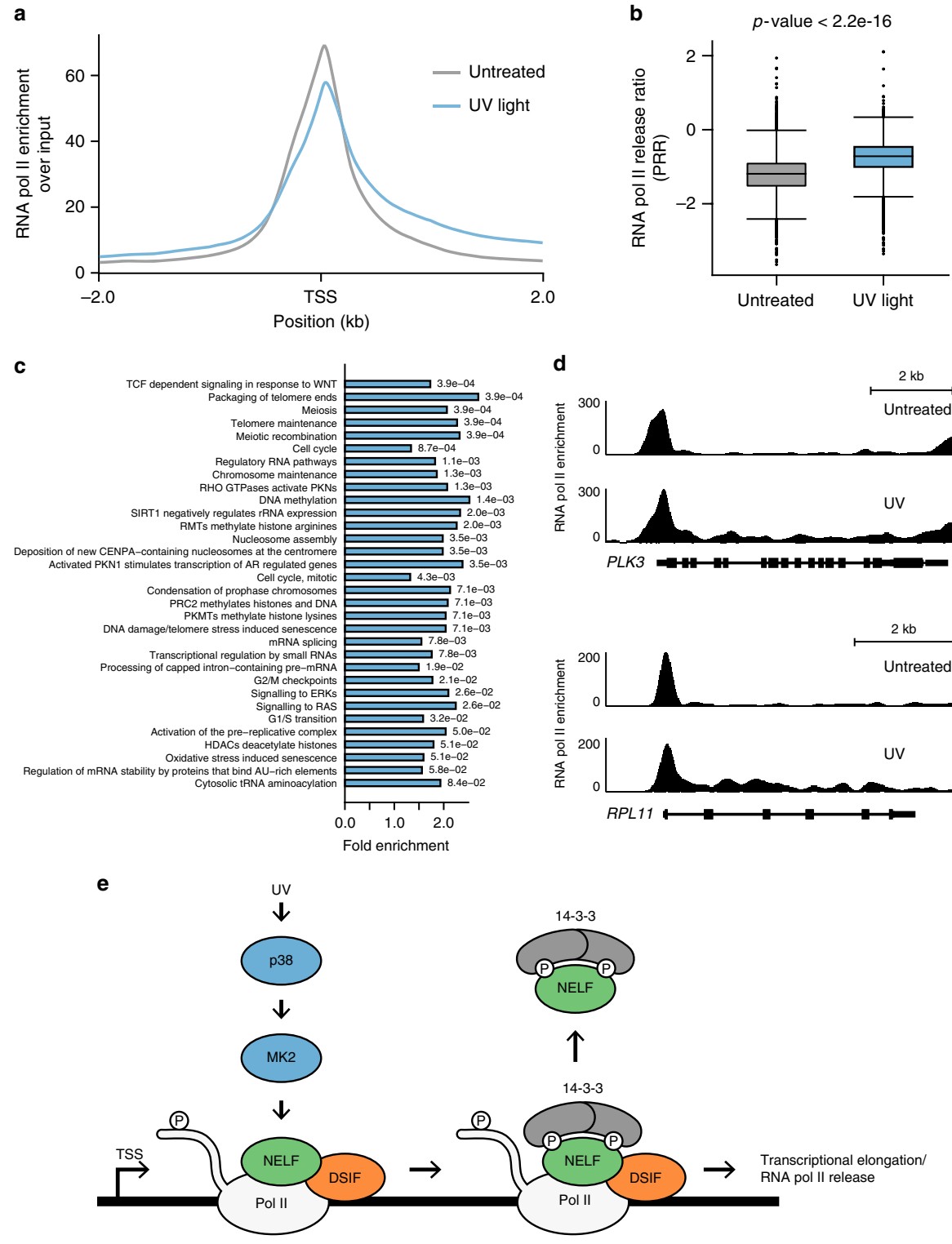

pathway[49]: NELFE and NELFA, but not the other subunits of the complex, were shown to be recruited to DSBs to repress transcription and promote DNA repair, suggesting that the NELF complex and transcriptional elongation is differentially regulated depending on the type of DNA damage.

Taken together, in this study we chart a map of phosphorylation events induced in human cells after irradiation with UV light and establish the dependencies of this phosphorylation on the canonical DNA damage signaling and the p38-MK2 signaling axis. The provided datasets of UV-light-induced phosphorylation sites and p38-dependent 14-3-3 interactions will enable further studies focusing on the functions of the p38-MK2 pathway in the regulation of different RNA metabolic processes after UV light.

## Methods

**Cell culture.** U2OS, HEK293T, RPE-1, and HaCaT cells were obtained from ATCC or DSMZ and cultured in Dulbecco's Modified Eagle Medium supplemented with 10% fetal bovine serum, L-glutamine, penicillin, and streptomycin. Cells were routinely tested for mycoplasma infection with a PCR-based method. For SILAC labeling, cells were cultured in media containing either L-arginine and L-lysine, L-arginine [13C6] and L-lysine [2H4] or L-arginine [13C615N4] and L-lysine [13C6-15N2] (Cambridge Isotope Laboratories)[50]. All cells were cultured at 37 °C in a humidified incubator containing 5% $CO_2$. Cells were transfected with siRNAs using Lipofectamine RNAiMAX (Life Technologies) according to the manufacturer's instructions. The complete list of siRNA sequences and oligos used in this study can be found in Supplementary Tables 1 and 2, respectively.

**Cell lysis for phosphoproteomics.** Cells were pretreated with 10 μM of p38 inhibitor SB203580 (Selleckchem) or MK2/3/5 inhibitor PF-3644022 (Sigma-Aldrich) for 1 h before irradiation with UV light (40 J/m²). After 1 h recovery period, cells were washed with ice-cold phosphate-buffered saline (PBS). Cells were lysed in modified RIPA buffer (50 mM Tris pH 7.5, 150 mM NaCl, 1 mM EDTA, 1% NP-40, 0.1% sodium deoxycholate) supplemented with protease inhibitors (Complete protease inhibitor cocktail tablets, Roche Diagnostics), 1 mM sodium orthovanadate, 5 mM β-glycerophosphate, and 5 mM sodium flouride (all from Sigma). Subsequently, lysates were cleared by centrifugation at $16,000 \times g$ for 15 min and protein concentrations were estimated using QuickStart Bradford Protein assay (BioRad).

**MS sample preparation.** Phosphoproteome analysis was performed exactly as described previously[51]. Briefly, proteins were precipitated in fourfold excess of ice-cold acetone and subsequently re-dissolved in denaturation buffer (6 M urea, 2 M thiourea in 10 mM HEPES pH 8.0). Cysteines were reduced with 1 mM dithiothreitol (DTT) and alkylated with 5.5 mM chloroacetamide. Proteins were digested with endoproteinase Lys-C (Wako Chemicals) and sequencing grade-modified trypsin (Sigma). Protease digestion was stopped by addition of trifluoroacetic acid to 0.5% and precipitates were removed by centrifugation. Peptides were purified using reversed-phase Sep-Pak C18 cartridges (Waters) and eluted in 50% acetonitrile. For the enrichment of phosphorylated peptides, 5 mg of peptides in binding buffer (50% acetonitrile, 6% trifluoroacetic acid in $H_2O$) were incubated with 10 mg of $TiO_2$ spheres (GL Sciences) for 1 h. The beads were washed twice in binding buffer and subsequently peptides were eluted using elution buffer (10% $NH_4OH$, 25% acetonitrile in $H_2O$). The eluates were concentrated to remove $NH_4OH$ and peptides were fractionated in six fractions using micro-column-based strong-cation exchange chromatography[52] and desalted on reversed-phase C18 StageTips[53].

**MS analysis.** Peptide fractions were analyzed on a quadrupole Orbitrap mass spectrometer (Q Exactive or Q Exactive Plus, Thermo Scientific) equipped with a UHPLC system (EASY-nLC 1000, Thermo Scientific) as described[54,55]. Peptide samples were loaded onto C18 reversed-phase columns (15 cm length, 75 μm inner diameter, 1.9 μm bead size) and eluted with a linear gradient from 8 to 40% acetonitrile containing 0.1% formic acid in 2 h. The mass spectrometer was operated in data-dependent mode, automatically switching between MS and MS² acquisition. Survey full scan MS spectra (m/z 300–1700) were acquired in the Orbitrap. The 10 most intense ions were sequentially isolated and fragmented by higher-energy C-trap dissociation (HCD)[56]. An ion selection threshold of 5,000 was used. Peptides with unassigned charge states, as well as with charge states less than + 2 were excluded from fragmentation. Fragment spectra were acquired in the Orbitrap mass analyzer.

**Peptide identification.** Raw data files were analyzed using MaxQuant (development version 1.5.2.8)[57]. Parent ion and MS² spectra were searched against a database containing 88,473 human protein sequences obtained from the UniProtKB released in December 2016 using Andromeda search engine[58]. Spectra were searched with a mass tolerance of 6 p.p.m. in MS mode, 20 p.p.m. in HCD MS² mode, strict trypsin specificity, and allowing up to three miscleavages. Cysteine carbamidomethylation was searched as a fixed modification, whereas protein N-terminal acetylation, methionine oxidation, and phosphorylation of serine, threonine, and tyrosine were searched as variable modifications. Site localization probabilities were determined by MaxQuant using the posttranslational modification scoring algorithm[57,59]. The dataset was filtered based on posterior error probability to arrive at a false discovery rate of below 1% estimated using a target-decoy approach[60]. Only phosphorylated peptides with a minimum score of 40 and delta score of 6 are reported and used for the analyses.

**Phosphorylation site occupancy analysis.** SILAC-labeled U2OS cells ectopically expressing GFP-NELFE (light) were mock treated or irradiated with UV light (medium and heavy). GFP-NELFE was enriched using GFP Trap agarose as described above, except that the washing was done with 8 M urea followed by washes with PBS. GFP-NELFE enriched from heavy-labeled SILAC condition was dephosphorylated with five units of Antarctic Phosphatase for 2 h at room temperature, whereas GFP-NELFE enriched from light- and medium-labeled cells was mock treated. GFP-NELFE-bound agarose beads from different SILAC conditions were washed three times with PBS and combined after the last wash. Raw data were analyzed with MaxQuant and site occupancies were calculated based on the ratio of the corresponding unmodified peptides in phosphatase treated and untreated samples using the following formula ((1 − 1/ratio) × 100%) as described previously[61].

**Extraction of chromatin-associated proteins.** Cells were washed with ice-cold PBS and collected using a cell scraper. Cells were lysed in Fractionation buffer A (10 mM HEPES pH 7.5, 10 mM KCl, 1.5 mM $MgCl_2$, 0.34 M glucose, 10% glycerol, 1 mM DTT, 0.1% Triton X-100) supplemented with protease and phosphatase inhibitors. Nuclei were pelleted by centrifugation at $1,300 \times g$ for 5 min and resuspended in Fractionation buffer B (3 mM EDTA, 0.2 mM EGTA, 1 mM DTT). After incubation, samples were centrifuged at $1700 \times g$ for 5 min and the chromatin pellet was dissolved in the modified RIPA buffer containing 450 mM NaCl. Digestion with Benzonase Nuclease was used to release chromatin bound proteins. The lysates were cleared by centrifugation at $16,000 \times g$ for 10 min.

**Computational analysis of proteomics data.** Statistical analysis was performed using the R software environment. Correlation coefficient ($\rho$) and significance were determined using Spearman's rank method. Differences in SILAC ratio variance were assessed using the Siegel–Tukey test. Statistical significance was calculated using Wilcoxon's rank-sum test. Functional protein interaction network analysis

**Fig. 7** UV light leads to an increase in transcriptional elongation. **a** Metagene analysis showing total RNA pol II occupancy measured by ChIP-seq in mock-treated U2OS cells and cells irradiated with UV light (40 J/m², 1 h recovery). All TSSs bound by RNA pol II in untreated cells and after UV light exposure were used for the analysis. Metagene analysis shows an average of two independent replicate ChIP-seq experiments. **b** Exposure of U2OS cells with UV light (40 J/m², 1 h recovery) promotes the release of RNA pol II into downstream regions of genes. The box plot shows the calculated RNA pol II release ratios (PRRs) in untreated cells and in cells irradiated with UV light. The average PRRs were calculated from two independent replicate experiments. The lower and upper hinges represent the first and third quartiles (25th and 75th percentiles, respectively). The line in the center of the box corresponds to the median of the data range. *P*-value was calculated using the Wilcoxon's rank-sum test with continuity correction. **c** REACTOME terms significantly enriched among genes with UV light upregulated PRRs. The bar plot shows significantly overrepresented REACTOME terms associated with genes containing upregulated PRR compared with all RNA pol II bound genes. The significance of the enrichment of a specific term was determined using a hypergeometric test. *P*-values were corrected for multiple hypotheses testing using the Benjamini and Hochberg FDR. **d** UCSC Genome Browser tracks displaying the density of RNA pol II around the *PLK3* and *RPL11* gene in untreated cells and after exposure of cells with UV light. **e** Model for the NELF complex regulation by p38-MK2. Exposure of human cells to UV light leads to rapid activation of p38 and its downstream effector kinase MK2. MK2 triggers widespread phosphorylation of RNA-binding proteins, including the NELF complex subunit NELFE. Site-specific NELFE phosphorylation on S115 induces its transient interaction with 14-3-3. NELFE phosphorylation leads to dissociation of NELFE from chromatin that is accompanied by RNA pol II elongation

was performed using interaction data from the STRING database[62]. Only interactions with a score > 0.7 are represented in the networks. Cytoscape version 3.1.1 was used for visualization of protein interaction networks[63].

**Cell viability assays**. Cell viability assay was performed using the CellTiter-Blue Cell Viability Assay (Promega) according to the manufacturer's instructions.

**Colony formation assays**. Cells were plated at low density 24 h after transfection with siRNA. Seventy-two hours post transfection, growth medium was aspirated, cells were washed with PBS, and irradiated with UV light using a custom-built UVC light irradiation chamber. Colonies were stained with 0.4% Coomassie Brilliant Blue in 20% ethanol solution and colonies containing at least 10 cells were counted 12–14 days after irradiation. The number of colonies formed after UV light was normalized to untreated control cells.

**Pull-down assays**. Cell lysates were prepared as described in the Cell lysis section. Twenty-five microliters of pre-equilibrated StrepTactin sepharose beads (IBA) or 20 μL of GFP Trap agarose (Chromotek) were added to the cleared lysate and incubated 1 h in the cold room on a rotation wheel. The beads were washed six times with modified RIPA buffer supplemented with protease and phosphatase inhibitors. In case of SILAC pull downs, beads from each SILAC condition were pooled after the fifth wash. Bound proteins were eluted in NuPAGE LDS Sample Buffer (Life Technologies) supplemented with 1 mM DTT, heated at 70 °C for 10 min, alkylated by addition of 5.5 mM chloroacetamide for 30 min, and loaded onto 4–12% gradient SDS-polycrylamide gel electrophoresis (PAGE) gels. Proteins were stained using the Colloidal Blue Staining Kit (Life Technologies) and digested in-gel using trypsin. Peptides were extracted from gel and desalted on reversed-phase C18 StageTips. Alternatively, proteins were transferred onto nitrocellulose membrane for western blotting.

**Peptide pull downs**. Biotinylated NELFE peptide (QPFQRSIpSADDLQE) was synthesized (GenScript) and bound to NeutrAvidin agarose. Peptide-bound agarose was mock-treated or subjected to dephosphorylation with λ-phosphatase. Subsequently, bound peptides were incubated with 1 μg of recombinant 14-3-3 for 3 h in the cold room. Pull downs were washed three times with buffer containing 1 × PBS, 300 mM NaCl, 0.1% Triton X-100, 2 mM DTT supplemented with protease and phosphatase inhibitors.

**Purification of GST-14-3-3 and GST pull-down assays**. *Escherichia coli* were transformed with a plasmid encoding GST-14-3-3 and protein expression was induced by addition of 1 mM isopropyl β-D-1-thiogalactopyranoside (IPTG) for 4 h at 25 °C. Cells were collected by centrifugation and lysed in Lysis buffer (50 mM Tris pH 7.5, 150 mM NaCl) supplemented with protease inhibitors, 200 μg/ml lysozyme, and 1 μg/ml Benzonase Nuclease. The lysates were incubated with Glutathione Sepharose 4B (GE Healthcare) for 3 h at 4 °C. Beads were washed six times with PBS and re-suspended in PBS supplemented with 5% glycerol. For GST pulldowns, 5 μg of GST-14-3-3 protein was incubated with 1 mg of protein extract for 2 h in the cold room. Pull downs were washed five times with modified RIPA buffer.

**Protein purification for crystallization studies**. The 14-3-3 epsilon was expressed in *E. coli* BL21 DE3 using the pET expression system. Cells were grown at 37 °C to an OD$_{600nm}$ of 0.5, followed by induction with 0.5 mM IPTG and further incubation at 25 °C for 16 h. Cells were lysed by sonication in buffer A (25 mM Tris, 200 mM NaCl, pH 8.5) and the supernatant was collected by centrifugation (15,000 × g, 4 °C, 40 min). The expressed protein was purified using TALON Metal Affinity Resin (Clontech), cleaved by TEV protease, and further purified by size exclusion chromatography (HiLoad 16/600 Superdex 75 column, GE Life Sciences) in 25 mM Tris, 200 mM NaCl, pH 8.5.

**Structure determination**. The 14-3-3 epsilon and synthesized NELFE phosphopeptide (QPFQRSI(p)SADDDLQE, GenScript) were mixed to a molar ratio of 1:5 for crystallization. The crystals of 14-3-3 epsilon in complex with NELFE phosphorylated peptide were obtained using 40% pentaerythritol propoxylate, 0.2 M sodium thiocyanate, 0.1 M HEPES pH 7.0, as a reservoir solution by the sitting-drop vapor diffusion method at 293 K. Diffraction data were collected at the Swiss Lightsource SLS, beam line PXIII, and processed with XDS[64]. The crystal structure was determined by molecular replacement using the 14-3-3 epsilon structure (PDB: 2BR9) as search model. Manual model building and refinement were done with Coot, CCP4 software suite, and Phenix[65–67]. The final statistics of the refined models are shown in Supplementary Table 3.

**SDS-PAGE and western blotting**. Proteins were resolved on 4-12% gradient SDS-PAGE gels (NuPAGE Bis-Tris Precast Gels, Life Technologies) and transferred onto nitrocellulose membranes. Membranes were blocked using 10% skimmed milk solution in PBS supplemented with 0.1% Tween-20. The list of antibodies used in this study and conditions can be found in Supplementary Table 4.

Secondary antibodies coupled to horseradish peroxidase (Jackson ImmunoResearch Laboratories) were used for immunodetection. The detection was performed with SuperSignal West Pico Chemiluminescent Substrate (Thermo Scientific). Uncropped scans of all western blots can be found in Supplementary Figure 8.

**In vitro kinase assay**. Ectopically expressed GFP-tagged NELFE was pulled down from total cell lysates using GFP Trap agarose. Beads were washed extensively with modified RIPA lysis buffer supplemented with 1 M NaCl and once with kinase buffer (25 mM HEPES pH 7.2, 25 mM MgCl, 2 mM DTT). Reactions were initiated by adding 400 ng recombinant MK2 (Abcam) and 25 mM ATP to each sample and then incubated for 30 min at 30 °C with gentle shaking. Samples were then resolved on SDS–PAGE and digested in-gel using trypsin. Peptides were extracted from the gel and peptide fractions were analyzed by LC-MS/MS.

**ChIP sequencing**. ChIP was done according to the protocol described by Arrigoni et al.[68] Cells ($2 \times 10^7$) were crosslinked in growth medium containing 1% formaldehyde for 5 min at room temperature, neutralized with 0.125 M glycine, and washed twice with PBS. Cell pellet was resuspended in Farnham buffer (5 mM PIPES pH 8.0, 85 mM KCl, 0.5% IGEPAL CA-630) and incubated for 15 min at 4 °C with rotation. Cells were sonicated in 1 ml AFA tubes (Covaris, 520080) using a Covaris S220 focused ultrasonicator (duty factor: 2%, cycles/burst: 200, intensity: 2, water temperature 4 °C). The sonication was stopped when more than 70% of nuclei were isolated (240 s). NEXON-isolated nuclei were washed twice with Farnham buffer and resuspended in 1 ml shearing buffer (10 mM Tris-HCl pH 8.0, 0.1% SDS, 1 mM EDTA) supplemented with inhibitors and sheared for 7 min to a fragment size distribution of 100–800 bp (Covaris S220 focused ultrasonicator; duty factor: 5%, cycles/burst: 200, intensity: 4, water temperature: 4 °C). Precleared chromatin (150 μg) was incubated overnight at 4 °C with 5 μg of RNA pol II antibody followed by 3 h incubation with 20 μl Dynabeads Protein G (Invitrogen). Beads were washed twice with Wash buffer 1 (10 mM Tris-HCl pH 8.0, 100 mM NaCl, 1 mM EDTA, 0.5 mM EGTA, 0.1% sodium deoxycholate, 0.5% N-Lauroylsarcosine), twice with Wash buffer 2 (0.25 M LiCl, 1% IGEPAL CA-630, 1% sodium deoxycholate, 1 mM EDTA) and the bound chromatin was eluted in 1% SDS, 0.1 M NaHCO₃. Crosslinks were reversed by incubation at 65 °C overnight with gentle shaking. Subsequently, chromatin was incubated with RNase A (0.2 mg/ml) for 30 min at 37 °C and then with proteinase K (50 μg/ml) for 3 h at 55 °C. DNA was purified by phenol–chloroform extraction followed by ethanol precipitation and recovered in 30 μl (IP samples) or 50 μl (inputs) RNase-free water. Real-time PCR on the ChIP-ed material was performed using SYBR Green (ABI).

**Next-generation sequencing data analysis**. The samples were sequenced on an Illumina Nextseq with 67 bp in length. Reads were mapped against GCRh37 with Bowtie 2 (version 2.2.9, -N 0 -L 32 -fr --local --maxins 1000 --minins 0). Post-processing was done using SAM tools (version 1.3.1). Peaks were called using MACS2 (version 2.1.1) with default parameters for Human. Afterwards a DiffBind (version 2.0.6 with DESeq2 1.12.4) analysis was performed to detect differentially bound regions for the control condition versus the UV condition. The peaks used for the analysis were the union of intersected peaks per condition. PRRs were calculated as follows: for each gene the TSS region was defined as – 300 bp upstream to 1000 bp downstream and the downstream region defined as 1000 bp downstream to 3 kb downstream (2 kb length). The PRR ratio was calculated as the $\log_2$ ratio between the enrichment in the downstream region toward the enrichment in the TSS. Only genes with more than 3 kb in length were considered for the analysis. The PRRs were filtered based on the highest signal on the TSS in the untreated condition. GRO-Seq data were obtained from GSE91011 for untreated (GSM2419224) and 2 h UV treatment condition (GSM2419225). The raw data were processed the same way as the RNA pol II ChIP-seq data including the calculation of the PRRs. The signal at the TSS was required to have a coverage of 1 RPKM in untreated and 2 h UV treatment condition to be included in the analysis. GO term analysis was performed using ClusterProfiler (version 3.0.5, minGSize = 3, maxGSize = 1000) and the Reactome analysis using ReactomePA (version 1.16.2, minGSize = 10) after obtaining EntrezIDs using biomaRt[69–71]. The background used for the analysis were all RNA pol II-bound genes identified by ChiP-seq.

**Data availability**. The MS proteomics data were deposited to the ProteomeXchange Consortium (http://proteomecentral.proteomexchange.org) via the PRIDE partner repository[72] with the dataset identifier PXD004255 [https://www.ebi.ac.uk/pride/archive/projects/PXD004255]. The genomics data were deposited in NCBI's Gene Expression Omnibus[73] (https://www.ncbi.nlm.nih.gov/geo/) and are accessible through GEO Series accession number GSE100580 [https://www.ncbi.nlm.nih.gov/geo/query/acc.cgi?acc=GSE100580]. The crystal structure of 14-3-3 epsilon in a complex with NELFE phospho-peptide reported in this study is deposited in PDB with the accession code 6EIH [http://www.rcsb.org/structure/6EIH]. All other data supporting the findings of this study are available from the corresponding author on reasonable request.

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

## Acknowledgements

We acknowledge the Paul Scherrer Institut, Villigen, Switzerland, for provision of synchrotron radiation beam time at the beamlines PX and PXIII of the SLS, and thank Meitian Wang and Vincent Olieric for assistance. We acknowledge funding from the European Community's Seventh Framework Programme (FP7/2007-2013) under BioStruct-X (grant agreement number 283570). The research in P.B.'s group is supported by the German Research Foundation (Emmy Noether Program, BE 5342/1-1 and SFB 1177 on Selective Autophagy) and the Marie Curie Career Integration Grant from the European Commission (630763). S.A.W. is supported by the LOEWE program Ubiquitin Networks (Ub-Net) of the State of Hesse (Germany), the Else Kröner-Fresenius-Stiftung (2015_A124), and the Else Kröner-Forschungskolleg Frankfurt. The research in M.A.'s group is supported by the Leibniz Award (to Ivan Dikic), the Cluster of Excellence "Macromolecular Complexes" (project EXC115), and the Volkswagen Stiftung. The Novo Nordisk Foundation Center for Protein Research is supported financially by the Novo Nordisk Foundation (grant agreement NNF14CC0001). We thank Anja Freiwald for assistance with mass spectrometry analysis. Support by the IMB genomics and bioinformatics core facility is gratefully acknowledged.

## Author contribution

P.B. and S.A.W. designed and supervised the research. M.E.B., A.V. and T.J. conducted experiments. S.K.S. helped with ChIP-seq experiments. M.A. performed crystallization studies. S.A.W. and P.B. analyzed the proteomics data. N.K. analyzed the genomics data. M.A.X.T. and S.B.-J. performed colony formation assays. N.M. and C.C. contributed ideas in the initial stage of the project. P.B. and S.A.W. wrote the manuscript. All authors read and commented on the manuscript.

## Additional information

**Competing interests:** The authors declare no competing interests.

