## [Peer Review File · Nature Communications]

Reviewers' comments:

Reviewer #1 (Remarks to the Author):

Borisova et al., submitted to Nature Communications

This is a very interesting study providing a comprehensive analysis of the phosphorylation events occurring upon UV irradiation. The authors provide evidence that, besides the ATR-driven signaling cascades, p38/MK2 activity represents a major fraction of the protein phosphorylations upon irradiation. Moreover, they identify a transcriptional elongation factor, NELFE, as an important target of MK2 in response to UV irradiation. Phosphorylated NELFE associates with 14-3-3 proteins and falls off the chromatin. This correlates with the increased RNA Polymerase II release from transcriptional start sites in irradiated cells.

I have only a few suggestions to further improve this paper. However, since it comprises already major advances in our understanding of the UV response, I wouldn't find it mandatory to address all of them. The authors might want to consider some of the suggestions to get an even more complete picture.

1. Most of the studies were performed using U2OS cells. This is fine, since this cell line is very widely studied regarding its DNA damage response, allowing comparison with other studies. However, U2OS cells are derived from an osteosarcoma, i. e. from a tissue that rarely if ever is exposed to UV irradiation under physiological circumstances. It would therefore be desirable to perform a few key experiments on cells that are closer to naturally UV-exposed tissue, for instance keratinocytes. Real skin would be ideal, but HaCat cells would work to get an idea of how universal the phosphorylations and the role of NELFE in UV-response really are.

2. Just to clarify: The cell line under study should be clearly indicated in each figure panel, either in the figure or in the legend to each panel.

3. Fig. 6b. The text says that the chromatin associations of the indicated proteins did NOT depend on p38. Thus, is the labeling erroneous? Should the legend to the yellow columns read: mean p38i+UV/untreated?

4. Fig. 6d, e: If feasible, it would be preferable to see the total proteins and the chromatin-associated proteins side-by-side in the same experiments.

5. Fig. 6f: The authors nicely show that NELFE is required for cell survival upon UV irradiation. But is the phosphorylation of NELFE required for survival, too? The authors could try to address this by doing a rescue experiment: first knocking down NELFE with the siRNA targeting its 3'UTR, then overexpressing NELFE from a plasmid, and then trying the same with the S115 mutant NELFE. Does the wild-type rescue, whereas the mutant does not?

6. Fig. 7: Is the increase in RNA pol II release actually a function of NELFE? Does it still respond to UV when NELFE is removed by knockdown? This would be of interest to validate the model shown in panel e.

Reviewer #2 (Remarks to the Author):

In this work, the authors demonstrated the significance of UV damage-induced phosphorylation of RNA-binding proteins. The authors showed that the NELF complex is a novel substrate of MK2 and that its phosphorylation and dissociation from chromatin by plays an important role in UV damage response. Many of the arguments are supported well by a number of experiments presented, and the findings of this work are potentially valuable to better understanding the complex mechanism of UV damage response. However, there are a few critical points that should be addressed.

(1) The authors have not addressed clearly whether, and how, MK2-induced phosphorylation of NELF and the resulting transcriptional change contributes to UV damage response. Their causal

relationship can be tested, for example, by re-introducing a NELF-E S115A mutant to NELF-E knockdown cells.

(2) The authors found that, upon UV treatment, the Pol II release ratio (PRR) is upregulated in 2123 genes, including those involved in telomere maintenance, RNA metabolism, and cell cycle. Does this change lead to changes in mRNA levels of these genes? Is NELF involved in the change of PRR upon UV treatment?

(3) The interaction between endogenous NELF and 14-3-3 proteins should be tested under UV-treated and control conditions.

Reviewer #3 (Remarks to the Author):

The authors study the response to UV radiation on a proteome wide scale using quantitative phosphoproteomics. They find a dependency on the ATM/ATR or p38 map kinase pathway, and further using pulldown experiments identify RNA binding proteins to be substrates, and 14-3-3 as readouts of p38-MK2 dependent phosphorylation. Finally, they establish a causal link between NELF complex subunit NELFE phosphorylation and transcriptional elongation.

The study is very comprehensive and technically advanced. I spend substantial time going through everything, and honestly have to say that I have no critical comments regarding the technical execution and the deductions drawn from the experiments. Thus as a very rare case I enthusiastically support the publication of the study essentially without any suggestions for revisions.

The only suggestion I would like to make, is to include either as a main figure or as a supplement as schematic figure illustrating all the technologies used and findings and conclusions from each type of experiment. Given the scope of the study this would greatly facilitate for the readers to not only understand the biological conclusion, but to also get better insight into how different technologies can synergize to answer biological questions.

Reviewers' comments:

Reviewer #1 (Remarks to the Author):

Borisova et al., submitted to Nature Communications

This is a very interesting study providing a comprehensive analysis of the phosphorylation events occurring upon UV irradiation. The authors provide evidence that, besides the ATR-driven signaling cascades, p38/MK2 activity represents a major fraction of the protein phosphorylations upon irradiation. Moreover, they identify a transcriptional elongation factor, NELFE, as an important target of MK2 in response to UV irradiation. Phosphorylated NELFE associates with 14-3-3 proteins and falls off the chromatin. This correlates with the increased RNA Polymerase II release from transcriptional start sites in irradiated cells.

I have only a few suggestions to further improve this paper. However, since it comprises already major advances in our understanding of the UV response, I wouldn't find it mandatory to address all of them. The authors might want to consider some of the suggestions to get an even more complete picture.

We thank the reviewer for the positive and constructive comments.

1. Most of the studies were performed using U2OS cells. This is fine, since this cell line is very widely studied regarding its DNA damage response, allowing comparison with other studies. However, U2OS cells are derived from an osteosarcoma, i. e. from a tissue that rarely if ever is exposed to UV irradiation under physiological circumstances. It would therefore be desirable to perform a few key experiments on cells that are closer to naturally UV-exposed tissue, for instance keratinocytes. Real skin would be ideal, but HaCat cells would work to get an idea of how universal the phosphorylations and the role of NELFE in UV-response really are.

We have now repeated the key experiments in the keratinocyte cell line HaCaT to show that the described UV light-induced pathway is also present in these cells. In particular, we showed that NELFE and 14-3-3 interact in HaCaT cells and that interaction is induced after UV light and dependent on p38 activity (Figure S4a). In addition, we showed that NELFE dissociates from chromatin in HaCaT cells after UV and in a p38-dependent manner (Figure S6e). We could also show the interaction between NELFE and 14-3-3 in HEK293T and RPE-1 cells (Figure S4a).

2. Just to clarify: The cell line under study should be clearly indicated in each figure panel, either in the figure or in the legend to each panel.

We have now added in all figure legends the information about the cell line being studied.

3. Fig. 6b. The text says that the chromatin associations of the indicated proteins did NOT depend on p38. Thus, is the labeling erroneous? Should the legend to the yellow columns read: mean p38i+UV/untreated?

We thank the reviewer for pointing this out. Indeed, the labeling of the yellow bars should be p38i + UV/untreated. We corrected this mistake in the Figure 6b, c and Figure S7a of the revised manuscript.

4. Fig. 6d, e: If feasible, it would be preferable to see the total proteins and the chromatin-associated proteins side-by-side in the same experiments.

We have added now also the protein amounts in total cell lysates obtained after lysis with modified RIPA buffer. These data are shown in Figure S6c and 6d.

5. Fig. 6f: The authors nicely show that NELFE is required for cell survival upon UV irradiation. But is the phosphorylation of NELFE required for survival, too? The authors could try to address this by doing a rescue experiment: first knocking down NELFE with the siRNA targeting its 3'UTR, then overexpressing NELFE from a plasmid, and then trying the same with the S115 mutant NELFE. Does the wild-type rescue, whereas the mutant does not?

We tried to directly investigate whether phosphorylation of NELFE is important for cellular survival after UV light as suggested by the reviewer. We were able to use a siRNA targeting the 3'-UTR region of NELFE to knockdown endogenous NELFE and to reconstitute these cells with N-terminally tagged Strep-Flag or GFP-tagged NELFE wild type or S115A mutant. However, we could not see a significant rescue of the survival after reconstitution of knockdown cells with wild type NELFE. This made it impossible for us to test the effect of the phospho mutant of NELFE in this assay. The functional NELF complex is composed of four protein subunits, NELFA, NELFB, NELFC or D and NELFE (PMID: 12612062). It is possible that the levels of the ectopically expressed NELFE differed from the levels of the endogenous NELFE in cells and hampered the functional reconstitution of the functional multi-subunit NELF complex. Although we tested two different protein tags, we cannot fully exclude that the tagged NELFE is not able to correctly incorporate into the multi-subunit NELF complex and thus not able to reconstitute a functional NELF complex in cells depleted of endogenous NELFE. To reflect that we were not directly able to link the NELFE phosphorylation to cell survival and transcriptional elongation after UV light exposure, we state in the abstract, results and discussion that the release of RNA pol II correlates with/or is accompanied by the dissociation of NELFE from chromatin and that other factors besides the NELF complex might be involved in regulation of transcriptional elongation after UV light.

6. Fig. 7: Is the increase in RNA pol II release actually a function of NELFE? Does it still respond to UV when NELFE is removed by knockdown? This would be of interest to validate the model shown in panel e.

In support of our findings, it has been recently shown that knockdown of the NELFE subunit of the NELF complex can induce global RNA pol II elongation in unstressed cancer and primary cells (PMID: 27256882, PMID: 28868519). We tried to perform RNA pol II ChIP-sequencing in unstressed keratinocyte cells (HaCaT) after NELFE knockdown to test whether release of RNA pol II happens also in non-irradiated HaCaT cells upon knockdown of NELFE. Due to technical difficulties in the experiment that manifested in poor enrichment of RNA pol II at transcription start sites in the ChIP-seq, we could not draw any valid conclusions. We added in the discussion section of the revised manuscript a sentence stating that NELFE knockdown has been previously shown to be sufficient to promote RNA pol II elongation. In addition, in the manuscript we write that RNA pol II release after UV light is correlated with or accompanied by the NELF complex removal from chromatin. We also state that it is possible that p38-dependent regulation of additional proteins involved in RNA pol II pausing such as TRIM28 and LARP7, which we also found to be phosphorylated by p38, contribute jointly to achieve RNA pol II elongation in response to UV light.

Reviewer #2 (Remarks to the Author):

In this work, the authors demonstrated the significance of UV damage-induced phosphorylation of RNA-binding proteins. The authors showed that the NELF complex is a novel substrate of MK2 and that its phosphorylation and dissociation from chromatin by plays an important role in UV damage response. Many of the arguments are supported well by a number of experiments presented, and the findings of this work are potentially valuable to better understanding the complex mechanism of UV damage response. However, there are a few critical points that should be addressed.

We thank the reviewer for the positive and constructive comments.

(1) The authors have not addressed clearly whether, and how, MK2-induced phosphorylation of NELF and the resulting transcriptional change contributes to UV damage response. Their causal relationship can be tested, for example, by re-introducing a NELF-E S115A mutant to NELF-E knockdown cells.

We tried to directly investigate whether phosphorylation of NELFE is important for cellular survival after UV light as suggested by the reviewer. We were able to use a siRNA targeting the 3'-UTR region of NELFE to knockdown endogenous NELFE and to reconstitute these cells with N-terminally tagged Strep-Flag or GFP-tagged NELFE wild type or S115A mutant. However, we could not see a significant rescue of the survival after reconstitution of knockdown cells with wild type NELFE. This made it

impossible for us to test the effect of the phospho mutant of NELFE in this assay. The functional NELF complex is composed of four protein subunits, NELFA, NELFB, NELFC or D and NELFE (PMID: 12612062). It is possible that the levels of the ectopically expressed NELFE differed from the levels of the endogenous NELFE in cells and hampered the functional reconstitution of the functional multi-subunit NELF complex. Although we tested two different protein tags, we cannot fully exclude that the tagged NELFE is not able to correctly incorporate into the multi-subunit NELF complex and thus not able to reconstitute a functional NELF complex in cells depleted of endogenous NELFE. To reflect that we were not directly able to link the NELFE phosphorylation to cell survival after UV light exposure, we state in the abstract, results and discussion that the release of RNA pol II correlates with/or is accompanied by the dissociation of NELFE from chromatin and that other factors besides NELF complex might be also involved in regulation of transcriptional elongation after UV light.

(2) The authors found that, upon UV treatment, the Pol II release ratio (PRR) is upregulated in 2123 genes, including those involved in telomere maintenance, RNA metabolism, and cell cycle. Does this change lead to changes in mRNA levels of these genes? Is NELF involved in the change of PRR upon UV treatment?

While this manuscript was under review, a study from another lab that used RNA pol II ChIP-seq and nascent RNA sequencing to monitor RNA pol II occupancy and changes in transcription after UV light was published. In agreement with our results presented in Figure 7 the authors reported that UV light leads to RNA pol II elongation, which is needed for enhanced DNA damage sensing and DNA repair (PMID: 29233992, Lavigne et al, Nature Communication). However, the authors do not address how these changes in RNA pol II elongation are achieved. In this study, the authors measured nascent transcripts and indeed showed that UV light triggers de novo transcription of essentially all genes (Figure 3 in Lavigne et al., Nature Communications). We have now added in the discussion the following sentence: "In agreement with our results, a recent study reported that UV light triggers RNA pol II elongation and transcription of active genes, which leads to enhanced DNA damage sensing by RNA pol II and DNA repair through the TC-NER pathway¹⁹".

In support of our findings, it has been recently shown that knockdown of the NELFE subunit of the NELF complex can induce global RNA pol II elongation in unstressed cancer and primary cells (PMID: 27256882, PMID: 28868519). We tried to perform RNA pol II ChIP-sequencing in unstressed keratinocyte cells (HaCaT) after NELFE knockdown to test whether release of RNA pol II happens also in non-irradiated HaCaT cells upon knockdown of NELFE. Due to technical difficulties in the experiment that manifested in poor enrichment of RNA pol II at transcription start sites in the ChIP-seq, we could not draw any conclusions. We added in the discussion section of the revised manuscript a sentence stating that NELF complex knockdown has been previously shown to be sufficient to promote RNA pol II elongation. In addition, in the manuscript we write that RNA pol II release after UV light is correlated with or accompanied by the NELF complex removal from chromatin. We also state that it is possible that p38-dependent regulation of additional proteins involved in RNA pol II pausing such as TRIM28 and LARP7, which we also found to be phosphorylated by p38, contribute jointly to achieve RNA pol II elongation in response to UV light.

(3) The interaction between endogenous NELF and 14-3-3 proteins should be tested under UV-treated and control conditions.

We showed that NELFE and 14-3-3 interact in human cells by different means: We pulled down tagged 14-3-3 and could detect endogenous NELFE from U2OS cell lysates (Figure 4b). We also pulled down tagged NELFE and detected endogenous 14-3-3 proteins by quantitative mass spectrometry (Figure 4a). We also purified recombinant GST-14-3-3 and used this as a bait to pull down endogenous NELFE from four different cancer and non-cancer-derived cell lines (U2OS, HEK293T, RPE-1, HaCaT) (Figure 4c and S4a). Furthermore, we have shown that the NELFE and 14-3-3 can interact directly in peptide pull downs by using the NELFE phosphopeptide and purified recombinant 14-3-3. We have also solved the crystal structure of the NELFE phosphopeptide with 14-3-3 (PDB ID 6EIH).

We tried to validate the interaction between NELFE and 14-3-3 in immunoprecipitation experiments where both proteins are present at endogenous level. We did not succeed due to technical challenges: Because of their molecular weight (NELFE 43 kDa; 14-3-3 29 kDa) are running in the gel at the same size as the heavy and light chain of the antibody that is used for the IP. We tried to do an IP both using NELFE and 14-3-3 antibodies, but in both cases, we have encountered the same problem of high background after trying to blot for either of the protein due to the interference of either light or heavy chain of the antibody. We tried to solve this problem by using conformation specific secondary antibodies that do not recognize the light and heavy antibody chain, but this did not solve the problem of high background.

Reviewer #3 (Remarks to the Author):

The authors study the response to UV radiation on a proteome wide scale using quantitative phosphoproteomics. They find a dependency on the ATM/ATR or p38 map kinase pathway, and further using pulldown experiments identify RNA binding proteins to be substrates, and 14-3-3 as readers of p38-MK2 dependent phosphorylation. Finally, they establish a causal link between NELF complex subunit NELFE phosphorylation and transcriptional elongation.

The study is very comprehensive and technically advanced. I spend substantial time going through everything, and honestly have to say that I have no critical comments regarding the technical execution and the deductions drawn from the experiments. Thus as a very rare case I enthusiastically support the publication of the study essentially without any suggestions for revisions.

The only suggestion I would like to make, is to include either as a main figure or as a supplement as schematic figure illustrating all the technologies used and findings and conclusions from each type of experiment. Given the scope of the study this would greatly facilitated for the readers to not only understand the biological conclusion, but to also get better insight into how different technologies can synergize to answer biological questions.

We thank the reviewer for the positive comments. We have now included a graphical summary of the methods employed and obtained results in the supplementary information (Supplementary figure 7f).

REVIEWERS' COMMENTS:

Reviewer #1 (Remarks to the Author):

My comments have been addressed in an appropriate way, and I suggest the acceptance of the paper.

Reviewer #2 (Remarks to the Author):

I think that the authors have answered almost all the comments in the response letter and have satisfactorily revised the manuscript. Although the authors could not present the data I requested, authors' explanation as to why the experiments were not successful is understandable.